# Dynamic Mode Decomposition-inspired Autoencoders for Reduced-order Modeling and Control of PDEs : Theory and Design

## Abstract

Modeling and controlling complex spatiotemporal dynamical systems driven by partial differential equations (PDEs) often necessitate dimensionality reduction techniques to construct lower-order models for computational efficiency. This paper studies a deep autoencoding learning method for modeling and controlling dynamical systems governed by spatiotemporal PDEs. We first analytically show that an optimization objective for learning a linear autoencoding reduced-order model can be formulated, yielding a solution that closely resembles the result obtained through the *dynamic mode decomposition with control* algorithm. Subsequently, we extend this linear autoencoding architecture to a deep autoencoding framework, enabling the development of a nonlinear reduced-order model. Furthermore, we leverage the learned reduced-order model to design controllers using stability-constrained deep neural networks. Empirical analyses are presented to validate the efficacy of our approach in both modeling and controlling spatiotemporal dynamical systems, exemplified through applications to reaction-diffusion systems and fluid flow systems.

## 1 Introduction

Performing high-fidelity simulations of physical systems governed by partial differential equations (PDEs) incurs substantial computational costs, rendering subsequent tasks, such as control, extremely challenging if not infeasible. To overcome the computational challenge, typically, reduced-order models (ROMs) are developed using dimensionality reduction techniques, enabling efficient simulation and control. For controlled dynamical systems, the reduced-order modeling approaches either combine analytical techniques with empirical approximation (Willcox & Peraire (2002)) or are purely data-driven (Juang & Pappa (1985); Juang et al. (1993); Proctor et al. (2016)). Among these, the dynamic mode decomposition (DMD) based methods have become widely popular in recent years due to a strong connection between DMD and Koopman operator theory (Rowley et al. (2009)). Another recent research trend involves utilizing deep neural networks (DNNs), particularly autoencoders, for modeling and control of high-dimensional dynamical systems (Lusch et al. (2018); Eivazi et al. (2020); Morton et al. (2018); Bounou et al. (2021); Chen et al. (2021)).

In this paper, our aim is to develop autoencoder-based ROMs for PDE-driven controlled dynamical systems and leverage these ROMs to learn control policies for the original systems. There are several viable approaches to constructing and training an autoencoder-based ROM for PDE-driven dynamical systems, as demonstrated in the aforementioned studies. However, a controller designed for the ROM is expected to perform well in the full system only if the ROM effectively captures the dynamic characteristics of the underlying system. DMD has become a widely used technique for extracting the dominant modes of underlying dynamics in a reduced-order model. This motivates us to develop an autoencoding framework for controlled dynamical systems drawing inspiration from the *dynamic mode decomposition with control* (DMDc) algorithm (Proctor et al. (2016)). We first formulate an objective function for data-driven model learning of controlled dynamical systems in a linear autoencoding configuration. We analytically show that the associated objective function encourages a linear ROM that closely resembles the lower-order model obtained using the DMDc algorithm. The linear autoencoding architecture is designed in such a way that its components can be replaced with DNNs and the corresponding objective function can be optimized by gradient descent to obtain a nonlinear ROM. A DNN-based nonlinear ROM provides more accurate predictions over

a longer temporal horizon, facilitating its integration into an offline control learning framework for the underlying system.

## 2 RELATED WORK

In recent years, deep learning has seen widespread application in scientific and engineering problems, including understanding complex spatiotemporal dynamics and solving associated computational tasks. The majority of the research in this area focuses only on the modeling and prediction of such complex dynamics using deep neural networks (DNNs) (Xingjian et al. (2015); Long et al. (2018); Raissi (2018); Seo et al. (2019); Ayed et al. (2019); Donà et al. (2020)) and has found application in several fields including fluid flow (Erichson et al. (2019); Eivazi et al. (2020); Srinivasan et al. (2019)), biochemical and electric power systems (Yeung et al. (2019)), climate and ocean systems (Scher (2018); Ren et al. (2021); Yang et al. (2017); De Bézenac et al. (2019)), and structural analysis Zhang et al. (2020), just to name a few. Encoder-decoder-based models, like the one utilized in our approach, stand as a prevalent choice among various deep prediction models. Vlachas et al. (2022) and Wiewel et al. (2020) combined encoder-decoder with recurrent network in latent space to accelerate long-range simulation. Wu et al. (2022) used an autoencoder with a latent evolution model, similar to ours; however, they considered inverse optimization only for static parameters. Kim et al. (2019) introduced a generative autoencoder with a latent space dynamic model to generate realistic fluid simulation from latent parameters. Lee & Carlberg (2020) used the learned latent representation from an autoencoder to form the trial basis for solving PDEs using Galerkin method.

A second line of research, though relatively less prevalent than modeling and prediction, is utilizing deep learning for controlling PDE-driven systems. Deep reinforcement learning (RL) is one of the approaches utilized to learn control policies for such systems (Rabault et al. (2019); Tang et al. (2020); Ma et al. (2018); Garnier et al. (2021); Beintema et al. (2020)). However, model-free RL methods require running numerical solvers in every iteration to provide feedback to the agents, which is computationally expensive. The same concern arises for the methods involving differentiable simulators as in Holl et al. (2020); Takahashi et al. (2021). In comparison, our method avoids the need for simulators during the learning as it learns from pre-collected data in an offline manner.

The alternative to model-free methods for control design takes the traditional approach: develop a model first and then use that to design controllers. Bieker et al. (2020) and Morton et al. (2018) used DNN-based model predictive control (MPC) framework, namely DeepMPC (Lenz et al. (2015)), in fluid flow control. Bieker et al. (2020) used a recurrent neural network to model the dynamics of only control-relevant quantities (i.e. lift and drag) of the system, which is then employed in an MPC framework for the flow control tasks. Morton et al. (2018) followed the method proposed by Takeishi et al. (2017) and used DNN-based embedding to first learn a linear reduced-order model in Koopman invariant subspace and then incorporate it in the MPC framework. Similar approaches have been applied for controlling other spatiotemporal dynamics like control from video input (Bounou et al. (2021)), automatic generation control in wind farms in the presence of dynamic wake effect (Chen et al. (2021)), and transient stabilization in power grids (Ping et al. (2021)). These model-based methods constrain the latent dynamic models to be linear and work well within a short time window. Khodkar et al. (2019) showed that the linear combination of a finite number of dynamic modes may not accurately represent the long-term nonlinear characteristics of complex dynamics and adding nonlinear forcing terms yields better prediction accuracy (Eivazi et al. (2020)). The linear ROMs need to be updated with online observations during operation for better prediction accuracy. Accordingly, the aforementioned model-based approaches utilize the MPC framework to optimize the control policy online using the updated dynamic model. Running online optimization at every step may not be computationally feasible in many scenarios. Conversely, we investigate if a nonlinear ROM provides a more accurate prediction over a longer time window so that an offline control learning method can be used.

## 3 PROBLEM AND PRELIMINARIES

### 3.1 PROBLEM STATEMENT

Consider a time-invariant controlled dynamical system driven by a PDE

$$\frac{\partial \mathcal{X}}{\partial t} = \mathcal{M}\left(\mathcal{X}, \frac{\partial \mathcal{X}}{\partial \zeta}, \frac{\partial^2 \mathcal{X}}{\partial \zeta^2}, \cdots, \mathcal{U}\right), \tag{1}$$

where $\mathcal{X}(\boldsymbol{\zeta}, t) \in \mathbb{R}$ and $\mathcal{U}(\boldsymbol{\zeta}, t) \in \mathbb{R}$ are the system state and the actuation (or control input), respectively, at location $\boldsymbol{\zeta}$ and time $t$. Space discretization of the state and actuation of (1) into $d_{\boldsymbol{x}}$ and $d_{\boldsymbol{u}}$ points, respectively, leads to a system of ordinary differential equations (ODEs) that can be written as

$$\frac{d\boldsymbol{x}}{dt} = f(\boldsymbol{x}, \boldsymbol{u}). \tag{2}$$

Here $\boldsymbol{x}(t) \in \mathbb{X} \subset \mathbb{R}^{d_{\boldsymbol{x}}}, d_{\boldsymbol{x}} >> 1$ and $\boldsymbol{u}(t) \in \mathbb{U} \subset \mathbb{R}^{d_{\boldsymbol{u}}}$ are the space-discretized state and actuation, respectively, at time $t$. Our objective is to learn a reduced-order model for this high-dimensional ($d_{\boldsymbol{x}} >> 1$) system of (2) and use that ROM to learn a feedback controller $\boldsymbol{u} = \pi(\boldsymbol{x})$ that stabilizes the system at a desired state. We consider a data-driven learning scenario and assume that we have observations from the system consisting of time series data $\boldsymbol{x}(t_i), i = 0, 1, \cdots, n$ subjected to random values of actuations $\boldsymbol{u}(t_i), i = 0, 1, \cdots, (n-1)$. Note, we use $v$ (in place of $v(t)$ for brevity) as notation for any continuous-time variable (e.g., system state, control input), whereas $v(t_i)$ is used to denote their discrete sample at time instance $t_i$. We further assume that the system we are aiming to stabilize at an equilibrium point is *locally stabilizable*, i.e., there exists a control policy such that the desired state is *asymptotically stable* for the closed-loop system. Readers are encouraged to reference the appendix A.1 for detailed formal definitions and constraints for stability.

## 3.2 DYNAMIC MODE DECOMPOSITION WITH CONTROL

DMD (Schmid (2010)) is a data-driven method that reconstructs the underlying dynamics using only a time series of snapshots from the system. DMDc (Proctor et al. (2016)) is an extension of DMD for dynamical systems with control. DMDc seeks best-fit linear operators $\boldsymbol{A}$ and $\boldsymbol{B}$ between successive observed states and the actuations:

$$\hat{\boldsymbol{x}}(t_{i+1}) = \boldsymbol{A}\boldsymbol{x}(t_i) + \boldsymbol{B}\boldsymbol{u}(t_i), \quad i = 0, 1, \cdots, n-1, \tag{3}$$

where $\hat{\boldsymbol{x}}(t)$ denotes an approximation of $\boldsymbol{x}(t)$, $\boldsymbol{A} \in \mathbb{R}^{d_{\boldsymbol{x}} \times d_{\boldsymbol{x}}}$, and $\boldsymbol{B} \in \mathbb{R}^{d_{\boldsymbol{x}} \times d_{\boldsymbol{u}}}$. Direct analysis of (3) could be computationally prohibitive for $d_{\boldsymbol{x}} >> 1$. DMDc leverages dimensionality reduction to compute a ROM

$$\boldsymbol{x}_{\text{R,DMDc}}(t_i) = \boldsymbol{E}_{\text{DMDc}}\boldsymbol{x}(t_i), \tag{4a}$$

$$\boldsymbol{x}_{\text{R,DMDc}}(t_{i+1}) = \boldsymbol{A}_{\text{R,DMDc}}\boldsymbol{x}_{\text{R,DMDc}}(t_i) + \boldsymbol{B}_{\text{R,DMDc}}\boldsymbol{u}(t_i), \quad i = 0, 1, \cdots, n-1, \tag{4b}$$

which retains the dominant dynamic modes of (3). Here, $\boldsymbol{x}_{\text{R,DMDc}}(t_i) \in \mathbb{R}^{r_{\boldsymbol{x}}}$ is the reduced state, where $r_{\boldsymbol{x}} << d_{\boldsymbol{x}}$, and $\boldsymbol{E}_{\text{DMDc}} \in \mathbb{R}^{r_{\boldsymbol{x}} \times d_{\boldsymbol{x}}}$, $\boldsymbol{A}_{\text{R,DMDc}} \in \mathbb{R}^{r_{\boldsymbol{x}} \times r_{\boldsymbol{x}}}$, $\boldsymbol{B}_{\text{R,DMDc}} \in \mathbb{R}^{r_{\boldsymbol{x}} \times d_{\boldsymbol{u}}}$. The full state is reconstructed from the reduced state using the transformation $\hat{\boldsymbol{x}}(t_i) = \boldsymbol{D}_{\text{DMDc}}\boldsymbol{x}_{\text{R,DMDc}}(t_i)$, where $\boldsymbol{D}_{\text{DMDc}} \in \mathbb{R}^{d_{\boldsymbol{x}} \times r_{\boldsymbol{x}}}$. DMDc computes truncated singular value decomposition (SVD) of the data matrices $\boldsymbol{Y} = [\boldsymbol{x}(t_1), \boldsymbol{x}(t_2), \cdots, \boldsymbol{x}(t_n)] \in \mathbb{R}^{d_{\boldsymbol{x}} \times n}$ and $\boldsymbol{\Omega} = [\boldsymbol{\omega}(t_0), \boldsymbol{\omega}(t_1), \cdots, \boldsymbol{\omega}(t_{n-1})] \in \mathbb{R}^{(d_{\boldsymbol{x}}+d_{\boldsymbol{u}}) \times n}$, $\boldsymbol{\omega}(t_i) = [\boldsymbol{x}(t_i)^{\top}, \boldsymbol{u}(t_i)^{\top}]^{\top} \in \mathbb{R}^{d_{\boldsymbol{x}}+d_{\boldsymbol{u}}}$ as follows:

$$\boldsymbol{Y} = \widehat{\boldsymbol{U}}_{\boldsymbol{Y}}\widehat{\boldsymbol{\Sigma}}_{\boldsymbol{Y}}\widehat{\boldsymbol{V}}_{\boldsymbol{Y}}^{\top}, \quad \boldsymbol{\Omega} = \widehat{\boldsymbol{U}}_{\boldsymbol{\Omega}}\widehat{\boldsymbol{\Sigma}}_{\boldsymbol{\Omega}}\widehat{\boldsymbol{V}}_{\boldsymbol{\Omega}}^{\top}, \tag{5}$$

where $\widehat{\boldsymbol{U}}_{\boldsymbol{Y}} \in \mathbb{R}^{d_{\boldsymbol{x}} \times r_{\boldsymbol{x}}}, \widehat{\boldsymbol{\Sigma}}_{\boldsymbol{Y}} \in \mathbb{R}^{r_{\boldsymbol{x}} \times r_{\boldsymbol{x}}}, \widehat{\boldsymbol{V}}_{\boldsymbol{Y}} \in \mathbb{R}^{n \times r_{\boldsymbol{x}}}, \widehat{\boldsymbol{U}}_{\boldsymbol{\Omega}} \in \mathbb{R}^{(d_{\boldsymbol{x}}+d_{\boldsymbol{u}}) \times r_{\boldsymbol{x}\boldsymbol{u}}}, \widehat{\boldsymbol{\Sigma}}_{\boldsymbol{\Omega}} \in \mathbb{R}^{r_{\boldsymbol{x}\boldsymbol{u}} \times r_{\boldsymbol{x}\boldsymbol{u}}}$, and $\widehat{\boldsymbol{V}}_{\boldsymbol{\Omega}} \in \mathbb{R}^{n \times r_{\boldsymbol{x}\boldsymbol{u}}}$. Here, $r_{\boldsymbol{x}} < \min(d_{\boldsymbol{x}}, n)$ and $r_{\boldsymbol{x}} < r_{\boldsymbol{x}\boldsymbol{u}} < \min(d_{\boldsymbol{x}} + d_{\boldsymbol{u}}, n)$ denote the truncation dimensions of SVDs. Utilizing the SVDs of (5), the parameters of the ROM (4) is obtained as

$$\boldsymbol{E}_{\text{DMDc}} = \widehat{\boldsymbol{U}}_{\boldsymbol{Y}}^{\top}, \quad \boldsymbol{D}_{\text{DMDc}} = \widehat{\boldsymbol{U}}_{\boldsymbol{Y}}, \tag{6a}$$

$$\boldsymbol{A}_{\text{R,DMDc}} = \widehat{\boldsymbol{U}}_{\boldsymbol{Y}}^{\top}\boldsymbol{Y}\widehat{\boldsymbol{V}}_{\boldsymbol{\Omega}}\widehat{\boldsymbol{\Sigma}}_{\boldsymbol{\Omega}}^{-1}\widehat{\boldsymbol{U}}_{\boldsymbol{\Omega},1}^{\top}\widehat{\boldsymbol{U}}_{\boldsymbol{Y}}, \quad \boldsymbol{B}_{\text{R,DMDc}} = \widehat{\boldsymbol{U}}_{\boldsymbol{Y}}^{\top}\boldsymbol{Y}\widehat{\boldsymbol{V}}_{\boldsymbol{\Omega}}\widehat{\boldsymbol{\Sigma}}_{\boldsymbol{\Omega}}^{-1}\widehat{\boldsymbol{U}}_{\boldsymbol{\Omega},2}^{\top}, \tag{6b}$$

where $\widehat{\boldsymbol{U}}_{\boldsymbol{\Omega},1} \in \mathbb{R}^{d_{\boldsymbol{x}} \times r_{\boldsymbol{x}\boldsymbol{u}}}, \widehat{\boldsymbol{U}}_{\boldsymbol{\Omega},2} \in \mathbb{R}^{d_{\boldsymbol{u}} \times r_{\boldsymbol{x}\boldsymbol{u}}}$, and $\widehat{\boldsymbol{U}}_{\boldsymbol{\Omega}}^{\top} = [\widehat{\boldsymbol{U}}_{\boldsymbol{\Omega},1}^{\top} \ \widehat{\boldsymbol{U}}_{\boldsymbol{\Omega},2}^{\top}]$.

# 4 METHOD

## 4.1 LEARNING A REDUCED ORDER MODEL

To develop a nonlinear ROM utilizing DNNs that effectively capture the underlying dynamics, we first investigate if we can obtain a linear ROM similar to DMDc, in a gradient descent arrangement. Specifically, we analyze optimization objectives that encourage a DMDc-like solution for a reduced-order modeling problem using linear networks (single layer without nonlinear activation). Consider the following reduced-order modeling problem

$$\boldsymbol{x}_{\text{R}}(t_i) = \boldsymbol{E}_{\boldsymbol{x}}\boldsymbol{x}(t_i), \ \boldsymbol{x}_{\text{R}}(t_{i+1}) = \boldsymbol{A}_{\text{R}}\boldsymbol{x}_{\text{R}}(t_i) + \boldsymbol{B}_{\text{R}}\boldsymbol{u}(t_i), \hat{\boldsymbol{x}}(t_i) = \boldsymbol{D}_{\boldsymbol{x}}\boldsymbol{x}_{\text{R}}(t_i), \ i = 0, 1, \cdots, n-1, \tag{7}$$

where the linear operators $\boldsymbol{E_x} \in \mathbb{R}^{r_x \times d_x}$ and $\boldsymbol{D_x} \in \mathbb{R}^{d_x \times r_x}$ projects and reconstructs back, respectively, the high-dimensional system state to and from a low-dimensional feature $\boldsymbol{x}_\mathrm{R} \in \mathbb{R}^{r_x}$. The linear operators $\boldsymbol{A}_\mathrm{R} \in \mathbb{R}^{r_x \times r_x}$ and $\boldsymbol{B}_\mathrm{R} \in \mathbb{R}^{r_x \times d_u}$ describe the relations between successive reduced states and actuations. We refer to this reduced-order model with linear networks as linear autoencoding ROM or LAROM. In the following, we first analyze the solution of the optimization objective of LAROM for a fixed *encoder* $\boldsymbol{E_x}$. Then we establish a connection between the solution of LAROM and the solution of DMDc, and further discuss the choice of the encoder to promote similarity between the two. Finally, we extend the linear model to a DNN-based model, which we refer to as DeepROM.

### 4.1.1 ANALYSIS OF THE LINEAR REDUCED-ORDER MODEL FOR A FIXED ENCODER

The DMDc algorithm essentially solves for $\widetilde{\boldsymbol{G}} \in \mathbb{R}^{r_x \times (d_x + d_u)}$ to minimize $\frac{1}{n} \sum_{i=0}^{n-1} \left\| \boldsymbol{E_x}\boldsymbol{x}(t_{i+1}) - \widetilde{\boldsymbol{G}}\boldsymbol{\omega}(t_i) \right\|^2$ for a fixed projection matrix $\boldsymbol{E_x} = \boldsymbol{E}_\mathrm{DMDc} = \widehat{\boldsymbol{U}}_{\boldsymbol{Y}}^\top$. Here, $\boldsymbol{\omega}(t_i)$ is the concatenated vector of state and actuation as defined in section 3.2. The optimal solution $\widetilde{\boldsymbol{G}}_\mathrm{opt}$ is then partitioned as $[\widetilde{\boldsymbol{A}} \ \widetilde{\boldsymbol{B}}]$ such that $\widetilde{\boldsymbol{A}} \in \mathbb{R}^{r_x \times d_x}, \widetilde{\boldsymbol{B}} \in \mathbb{R}^{r_x \times d_u}$. Finally, $\widetilde{\boldsymbol{A}}$ is post-multiplied with the reconstruction operator $\boldsymbol{D}_\mathrm{DMDc} = \widehat{\boldsymbol{U}}_{\boldsymbol{Y}}$ to get the ROM components $\boldsymbol{A}_{\mathrm{R},\mathrm{DMDc}}$ and $\boldsymbol{B}_{\mathrm{R},\mathrm{DMDc}}$. Details of this process along with the proofs are given in appendix B.5. Note, that the final step of this process (multiplication of the operators) is feasible only for the linear case, not in the case when the projection and reconstruction operators are nonlinear (e.g. DNNs). Therefore, we use an alternative formulation with the following results to design a loss function that encourages a DMDc-like solution for (7) and also offers dimensionality reduction when nonlinear components are used.

**Theorem 4.1.1.** *Consider the following objective function*

$$L_\mathrm{pred}(\boldsymbol{E_x}, \boldsymbol{G}) = \frac{1}{n} \sum_{i=0}^{n-1} \left\| \boldsymbol{E_x}\boldsymbol{x}(t_{i+1}) - \boldsymbol{G}\boldsymbol{E_{xu}}\boldsymbol{\omega}(t_i) \right\|^2, \tag{8}$$

*where* $\boldsymbol{G} = [\boldsymbol{A}_\mathrm{R} \ \boldsymbol{B}_\mathrm{R}] \in \mathbb{R}^{r_x \times (r_x + d_u)}, \boldsymbol{E_{xu}} = \begin{bmatrix} \boldsymbol{E_x} & \boldsymbol{0} \\ \boldsymbol{0} & \boldsymbol{I}_{d_u} \end{bmatrix} \in \mathbb{R}^{(r_x + d_u) \times (d_x + d_u)}, \boldsymbol{I}_{d_u}$ *being the identity matrix of order* $d_u$. *For any fixed matrix* $\boldsymbol{E_x}$, *the objective function* $L_\mathrm{pred}$ *is convex in the coefficients of* $\boldsymbol{G}$ *and attains its minimum for any* $\boldsymbol{G}$ *satisfying*

$$\boldsymbol{G}\boldsymbol{E_{xu}}\boldsymbol{\Omega}\boldsymbol{\Omega}^\top \boldsymbol{E_{xu}}^\top = \boldsymbol{E_x}\boldsymbol{Y}\boldsymbol{\Omega}^\top \boldsymbol{E_{xu}}^\top, \tag{9}$$

*where* $\boldsymbol{Y}$ *and* $\boldsymbol{\Omega}$ *are the data matrices as defined in section (3.2). If* $\boldsymbol{E_x}$ *has full rank* $r_x$, *and* $\boldsymbol{\Omega}\boldsymbol{\Omega}^\top$ *is non-singular, then* $L_\mathrm{pred}$ *is strictly convex and has a unique minimum for*

$$\boldsymbol{G} = [\boldsymbol{A}_\mathrm{R} \ \boldsymbol{B}_\mathrm{R}] = \boldsymbol{E_x}\boldsymbol{Y}\boldsymbol{\Omega}^\top \boldsymbol{E_{xu}}^\top (\boldsymbol{E_{xu}}\boldsymbol{\Omega}\boldsymbol{\Omega}^\top \boldsymbol{E_{xu}}^\top)^{-1}. \tag{10}$$

*Proof sketch.* For any fixed $\boldsymbol{E_x}$, the objective function of (8) can be written as $L_\mathrm{pred}(\boldsymbol{E_x}, \boldsymbol{G}) = \left\| \mathrm{vec}(\boldsymbol{E_x}\boldsymbol{Y}) - (\boldsymbol{\Omega}^\top \boldsymbol{E_{xu}}^\top \otimes \boldsymbol{I}_{r_x})\mathrm{vec}(\boldsymbol{G}) \right\|^2$, where $\otimes$ denotes the Kronecker product and $\mathrm{vec}(\cdot)$ denotes vectorization of a matrix. Optimizing this linear least-square problem, we get (9) and (10), given the stated conditions are satisfied. The complete proof is given in appendix B.1.

**Remark.** For a unique solution, we assume that $\boldsymbol{E_x}$ has full rank. The other scenario, i.e., $\boldsymbol{E_x}$ is rank-deficient suggests poor utilization of the hidden units of the model. In that case, the number of hidden units (which represents the dimension of the reduced state) should be decreased. The assumption that the covariance matrix $\boldsymbol{\Omega}\boldsymbol{\Omega}^\top$ is invertible can be ensured when $n \geq d_x + d_u$, by removing any linearly dependent features in system state and actuation. When $n < d_x + d_u$, the covariance matrix $\boldsymbol{\Omega}\boldsymbol{\Omega}^\top$ is not invertible. However, similar results can be obtained by adding $\ell_2$ regularization (for the coefficients/entries of $\boldsymbol{G}$) to the objective function. Proof of this is given in appendix B.4.

### 4.1.2 THE CONNECTION BETWEEN THE SOLUTIONS OF THE LINEAR AUTOENCODING MODEL AND DMDC

The connection between the ROM obtained by minimizing $L_\mathrm{pred}$ (for a fixed $\boldsymbol{E_x}$), i.e., (10) and the DMDc ROM of (6b) is not readily apparent. To interpret the connection, we formulate an alternative representation of (10) utilizing the SVD and the Moore-Penrose inverse of matrices. This alternative representation leads to the following result.

**Corollary 4.1.1.1.** *Consider the (full) SVD of the data matrix $\boldsymbol{\Omega}$ given by $\boldsymbol{\Omega} = \boldsymbol{U}_{\boldsymbol{\Omega}} \boldsymbol{\Sigma}_{\boldsymbol{\Omega}} \boldsymbol{V}_{\boldsymbol{\Omega}}^{\top}$, where $\boldsymbol{U}_{\boldsymbol{\Omega}} \in \mathbb{R}^{(d_{\boldsymbol{x}}+d_{\boldsymbol{u}}) \times (d_{\boldsymbol{x}}+d_{\boldsymbol{u}})}$, $\boldsymbol{\Sigma}_{\boldsymbol{\Omega}} \in \mathbb{R}^{(d_{\boldsymbol{x}}+d_{\boldsymbol{u}}) \times n}$, and $\boldsymbol{V}_{\boldsymbol{\Omega}} \in \mathbb{R}^{n \times n}$. If $\boldsymbol{E}_{\boldsymbol{x}} = \widehat{\boldsymbol{U}}_{\boldsymbol{Y}}^{\top}$ and $\boldsymbol{\Omega} \boldsymbol{\Omega}^{\top}$ is non-singular, then the solution for $\boldsymbol{G} = [\boldsymbol{A}_{\mathrm{R}} \quad \boldsymbol{B}_{\mathrm{R}}]$ corresponding to the unique minimum of $L_{pred}$ can be expressed as*

$$\boldsymbol{A}_R = \widehat{\boldsymbol{U}}_{\boldsymbol{Y}}^{\top} \boldsymbol{Y} \boldsymbol{V}_{\boldsymbol{\Omega}} \boldsymbol{\Sigma}^* \boldsymbol{U}_{\boldsymbol{\Omega},1}^{\top} \widehat{\boldsymbol{U}}_{\boldsymbol{Y}}, \quad \text{and} \quad \boldsymbol{B}_R = \widehat{\boldsymbol{U}}_{\boldsymbol{Y}}^{\top} \boldsymbol{Y} \boldsymbol{V}_{\boldsymbol{\Omega}} \boldsymbol{\Sigma}^* \boldsymbol{U}_{\boldsymbol{\Omega},2}^{\top}, \tag{11}$$

*where $[\boldsymbol{U}_{\boldsymbol{\Omega},1}^{\top} \quad \boldsymbol{U}_{\boldsymbol{\Omega},2}^{\top}] = \boldsymbol{U}_{\boldsymbol{\Omega}}^{\top}$ with $\boldsymbol{U}_{\boldsymbol{\Omega},1} \in \mathbb{R}^{d_{\boldsymbol{x}} \times (d_{\boldsymbol{x}}+d_{\boldsymbol{u}})}, \boldsymbol{U}_{\boldsymbol{\Omega},2} \in \mathbb{R}^{d_{\boldsymbol{u}} \times (d_{\boldsymbol{x}}+d_{\boldsymbol{u}})}$, and*
$\boldsymbol{\Sigma}^* = \lim_{\varepsilon \to 0} (\boldsymbol{\Sigma}_{\boldsymbol{\Omega}}^{\top} \boldsymbol{U}_{\boldsymbol{\Omega},1}^{\top} \widehat{\boldsymbol{U}}_{\boldsymbol{Y}} \widehat{\boldsymbol{U}}_{\boldsymbol{Y}}^{\top} \boldsymbol{U}_{\boldsymbol{\Omega},1} \boldsymbol{\Sigma}_{\boldsymbol{\Omega}} + \boldsymbol{\Sigma}_{\boldsymbol{\Omega}}^{\top} \boldsymbol{U}_{\boldsymbol{\Omega},2}^{\top} \boldsymbol{U}_{\boldsymbol{\Omega},2} \boldsymbol{\Sigma}_{\boldsymbol{\Omega}} + \varepsilon^2 \boldsymbol{I}_n)^{-1} \boldsymbol{\Sigma}_{\boldsymbol{\Omega}}^{\top}$.

*Proof sketch.* This can be derived by plugging $\boldsymbol{E}_{\boldsymbol{x}} = \widehat{\boldsymbol{U}}_{\boldsymbol{Y}}^{\top}$ into (10), and using the *SVD definition* and the *limit definition* (Albert (1972)) of the Moore-Penrose inverse. The complete proof is given in appendix B.3 that uses an alternative representation of (10) presented in appendix B.2.

**Remark.** It can be verified easily that if we use the truncated SVD (as defined by 5), instead of the full SVD, for $\boldsymbol{\Omega}$ in corollary 4.1.1.1, we get an approximation of (11):

$$\widehat{\boldsymbol{A}}_{\mathrm{R}} = \widehat{\boldsymbol{U}}_{\boldsymbol{Y}}^{\top} \boldsymbol{Y} \widehat{\boldsymbol{V}}_{\boldsymbol{\Omega}} \widehat{\boldsymbol{\Sigma}}^* \widehat{\boldsymbol{U}}_{\boldsymbol{\Omega},1}^{\top} \widehat{\boldsymbol{U}}_{\boldsymbol{Y}}, \quad \text{and} \quad \widehat{\boldsymbol{B}}_{\mathrm{R}} = \widehat{\boldsymbol{U}}_{\boldsymbol{Y}}^{\top} \boldsymbol{Y} \widehat{\boldsymbol{V}}_{\boldsymbol{\Omega}} \widehat{\boldsymbol{\Sigma}}^* \widehat{\boldsymbol{U}}_{\boldsymbol{\Omega},2}^{\top}, \tag{12}$$

where $\widehat{\boldsymbol{\Sigma}}^* = \lim_{\varepsilon \to 0} (\widehat{\boldsymbol{\Sigma}}_{\boldsymbol{\Omega}}^{\top} \widehat{\boldsymbol{U}}_{\boldsymbol{\Omega},1}^{\top} \widehat{\boldsymbol{U}}_{\boldsymbol{Y}} \widehat{\boldsymbol{U}}_{\boldsymbol{Y}}^{\top} \widehat{\boldsymbol{U}}_{\boldsymbol{\Omega},1} \widehat{\boldsymbol{\Sigma}}_{\boldsymbol{\Omega}} + \widehat{\boldsymbol{\Sigma}}_{\boldsymbol{\Omega}}^{\top} \widehat{\boldsymbol{U}}_{\boldsymbol{\Omega},2}^{\top} \widehat{\boldsymbol{U}}_{\boldsymbol{\Omega},2} \widehat{\boldsymbol{\Sigma}}_{\boldsymbol{\Omega}} + \varepsilon^2 \boldsymbol{I}_{r_{\boldsymbol{x}\boldsymbol{u}}})^{-1} \widehat{\boldsymbol{\Sigma}}_{\boldsymbol{\Omega}}^{\top}$. We can see that (12) has the same form as (6b), except $\widehat{\boldsymbol{\Sigma}}_{\boldsymbol{\Omega}}^{-1}$ is replaced with $\widehat{\boldsymbol{\Sigma}}^*$.

All the aforementioned results are derived for a fixed $\boldsymbol{E}_{\boldsymbol{x}}$ and the relation to the DMDc is specific to the case $\boldsymbol{E}_{\boldsymbol{x}} = \widehat{\boldsymbol{U}}_{\boldsymbol{Y}}^{\top}$. Note that the columns of the $\widehat{\boldsymbol{U}}_{\boldsymbol{Y}}$ are the left singular vectors, corresponding to the leading singular values, of $\boldsymbol{Y}$. Equivalently, those are also the eigenvectors, corresponding to the leading eigenvalues, of the covariance matrix $\boldsymbol{Y}\boldsymbol{Y}^{\top}$. $L_{\mathrm{pred}}$ alone does not constrain $\boldsymbol{E}_{\boldsymbol{x}}$ to take a similar form and we need another loss term to encourage such form for the encoder. To this end, we follow the work of Baldi & Hornik (1989) on the similarity between principle component analysis and linear autoencoders, optimized with the objective function: $L_{\mathrm{recon}}(\boldsymbol{E}_{\boldsymbol{x}}, \boldsymbol{D}_{\boldsymbol{x}}) = \frac{1}{n} \sum_{i=1}^{n} \left\| \boldsymbol{x}(t_i) - \boldsymbol{D}_{\boldsymbol{x}} \boldsymbol{E}_{\boldsymbol{x}} \boldsymbol{x}(t_i) \right\|^2$. They showed that all the critical points of $L_{\mathrm{recon}}$ correspond to projections onto subspaces associated with subsets of eigenvectors of the covariance matrix $\boldsymbol{Y}\boldsymbol{Y}^{\top}$. Moreover, $L_{\mathrm{recon}}$ has a unique global minimum corresponding to the first $r_{\boldsymbol{x}}$ (i.e., the desired dimension of the reduced state) number of eigenvectors of $\boldsymbol{Y}\boldsymbol{Y}^{\top}$, associated with the leading $r_{\boldsymbol{x}}$ eigenvalues. In other words, for any invertible matrix $\boldsymbol{C} \in \mathbb{R}^{r_{\boldsymbol{x}} \times r_{\boldsymbol{x}}}$, $\boldsymbol{D}_{\boldsymbol{x}} = \boldsymbol{U}_{r_{\boldsymbol{x}}} \boldsymbol{C}$ and $\boldsymbol{E}_{\boldsymbol{x}} = \boldsymbol{C}^{-1} \boldsymbol{U}_{r_{\boldsymbol{x}}}^{\top}$ globally minimizes $L_{\mathrm{recon}}$, where $\boldsymbol{U}_{r_{\boldsymbol{x}}}$ denotes the matrix containing leading $r_{\boldsymbol{x}}$ eigenvectors of $\boldsymbol{Y}\boldsymbol{Y}^{\top}$. Since the left singular vectors of $\boldsymbol{Y}$ are the eigenvectors of $\boldsymbol{Y}\boldsymbol{Y}^{\top}$, we have $\boldsymbol{U}_{r_{\boldsymbol{x}}} = \widehat{\boldsymbol{U}}_{\boldsymbol{Y}}$. Hence, we consider to utilize $L_{\mathrm{recon}}$ to promote learning an encoder $\boldsymbol{E}_{\boldsymbol{x}}$ in the form of $\boldsymbol{C}^{-1} \widehat{\boldsymbol{U}}_{\boldsymbol{Y}}^{\top}$. Accordingly, we propose to minimize the following objective function to encourage a DMDc-like solution for LAROM:

$$L(\boldsymbol{E}_{\boldsymbol{x}}, \boldsymbol{D}_{\boldsymbol{x}}, \boldsymbol{G}) = L_{\mathrm{pred}}(\boldsymbol{E}_{\boldsymbol{x}}, \boldsymbol{G}) + \beta_1 L_{\mathrm{recon}}(\boldsymbol{E}_{\boldsymbol{x}}, \boldsymbol{D}_{\boldsymbol{x}}), \tag{13}$$

where $\beta_1 > 0$ is a tunable hyperparameter.

### 4.1.3 EXTENDING THE LINEAR MODEL TO A DEEP MODEL

Here, we discuss the process of extending LAROM to a nonlinear reduced-order modeling framework. We replace all the trainable components of LAROM, i.e., $\boldsymbol{E}_{\boldsymbol{x}}, \boldsymbol{D}_{\boldsymbol{x}}$, and $\boldsymbol{G}$, with DNNs. Specifically, we use an encoding function or *encoder* $\mathcal{E}_{\boldsymbol{x}} : \mathbb{X} \to \mathbb{R}^{r_{\boldsymbol{x}}}$ and a decoding function or *decoder* $\mathcal{D}_{\boldsymbol{x}} : \mathbb{R}^{r_{\boldsymbol{x}}} \to \mathbb{X}$ to transform the high-dimensional system state to low-dimensional features and reconstruct it back, respectively, i.e., $\boldsymbol{x}_{\mathrm{R}} = \mathcal{E}_{\boldsymbol{x}}(\boldsymbol{x})$, $\hat{\boldsymbol{x}} = \mathcal{D}_{\boldsymbol{x}}(\boldsymbol{x}_{\mathrm{R}})$, where $\boldsymbol{x}_{\mathrm{R}} \in \mathbb{R}^{r_{\boldsymbol{x}}}$ denotes the reduced state, and $\hat{\boldsymbol{x}}$ is the reconstruction of $\boldsymbol{x}$. Unlike the linear case, we use an encoder $\mathcal{E}_{\boldsymbol{u}} : \mathbb{U} \to \mathbb{R}^{r_{\boldsymbol{u}}}, r_{\boldsymbol{u}} << d_{\boldsymbol{u}}$ for the actuation as well, in cases where the control space is also high-dimensional (for example, distributed control of spatiotemporal PDEs). The control encoder $\mathcal{E}_{\boldsymbol{u}}$ maps the high-dimensional actuation to a low-dimensional representation: $\boldsymbol{u}_{\mathrm{R}} = \mathcal{E}_{\boldsymbol{u}}(\boldsymbol{u})$, where $\boldsymbol{u}_{\mathrm{R}} \in \mathbb{R}^{r_{\boldsymbol{u}}}$ denotes the encoded actuation. The encoded state and control are then fed to another DNN that represents the reduced order dynamics $\frac{d\boldsymbol{x}_{\mathrm{R}}}{dt} = \mathcal{F}(\boldsymbol{x}_{\mathrm{R}}, \boldsymbol{u}_{\mathrm{R}})$, where $\mathcal{F} : \mathbb{R}^{r_{\boldsymbol{x}}} \times \mathbb{R}^{r_{\boldsymbol{u}}} \to \mathbb{R}^{r_{\boldsymbol{x}}}$. Given the current reduced state $\boldsymbol{x}_{\mathrm{R}}(t_i)$ and control input $\boldsymbol{u}_{\mathrm{R}}(t_i)$, the next reduced state $\boldsymbol{x}_{\mathrm{R}}(t_{i+1})$ can be computed by integrating $\mathcal{F}$ using a numerical integrator: $\boldsymbol{x}_{\mathrm{R}}(t_{i+1}) = \boldsymbol{x}_{\mathrm{R}}(t_i) + \int_{t_i}^{t_{i+1}} \mathcal{F}(\boldsymbol{x}_{\mathrm{R}}(t), \boldsymbol{u}_{\mathrm{R}}(t)) dt \stackrel{\triangle}{=} \mathcal{G}(\boldsymbol{x}_{\mathrm{R}}(t_i), \boldsymbol{u}_{\mathrm{R}}(t_i))$. We can say that $\mathcal{G}$ is the nonlinear counterpart of $\boldsymbol{G}$.

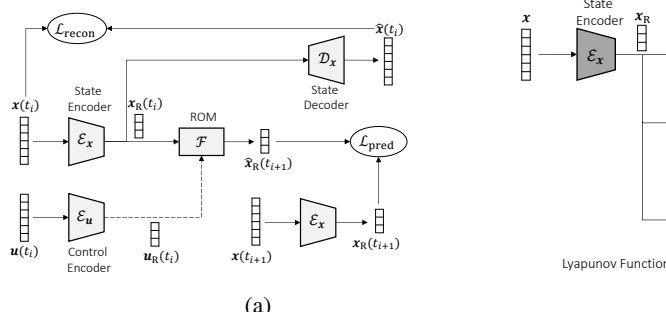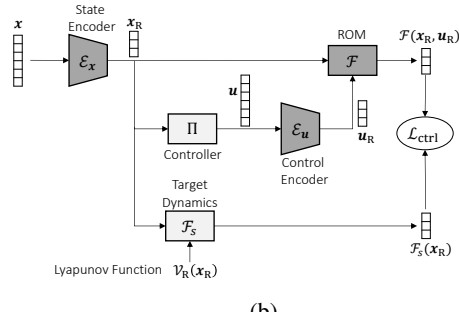

(a)                                      (b)

Figure 1: (a): Autoencoding architecture for reduced-order modeling. The state encoder $\mathcal{E}_{\boldsymbol{x}}$ and control encoder $\mathcal{E}_{\boldsymbol{u}}$ reduce the dimension of the state and actuation, respectively. The ROM $\mathcal{F}$ takes the current reduced state and actuation to predict the next reduced state, which is then uplifted to the full state by the state decoder $\mathcal{D}_{\boldsymbol{x}}$. All modules are trained together using a combined loss involving $\mathcal{L}_{\text{pred}}$ and $\mathcal{L}_{\text{recon}}$. The dashed arrow indicates that the $\mathcal{E}_{\boldsymbol{u}}$ is used only when $d_{\boldsymbol{u}} >> 1$; otherwise, the actuation is used as a direct input to ROM. (b): The control learning process. Given a reduced state, $\mathcal{F}_s$ predicts a target dynamics for the closed-loop system, and the controller $\Pi$ predicts an actuation to achieve that target. Both the modules are trained jointly using the loss function $\mathcal{L}_{\text{ctrl}}$. Parameters of the dark-shaded modules are kept fixed during this process.

Note, here the ROM is represented as a continuous-time dynamics, unlike the linear case where we used a discrete-time model. We use a discrete-time formulation for LAROM to establish its similarity with DMDc, which is formulated in discrete time. DeepROM can be formulated in a similar fashion as well. However, the specific control learning algorithm we used, which will be discussed in the next subsection, requires vector fields of the learned ROM for training. Therefore, we formulate the ROM in continuous time so that it provides the vector field $\mathcal{F}(\boldsymbol{x}_{\text{R}}, \boldsymbol{u}_{\text{R}})$ of the dynamics. In cases where only the prediction model is of interest and control learning is not required, a discrete-time formulation should be used for faster training of the ROM.

We train $\mathcal{E}_{\boldsymbol{x}}, \mathcal{E}_{\boldsymbol{u}}, \mathcal{D}_{\boldsymbol{x}}$, and $\mathcal{F}$ by minimizing the following loss function, analogous to (13),

$$\mathcal{L}(\mathcal{E}_{\boldsymbol{x}}, \mathcal{E}_{\boldsymbol{u}}, \mathcal{D}_{\boldsymbol{x}}, \mathcal{F}) = \mathcal{L}_{\text{pred}}(\mathcal{E}_{\boldsymbol{x}}, \mathcal{E}_{\boldsymbol{u}}, \mathcal{F}) + \beta_2 \mathcal{L}_{\text{recon}}(\mathcal{E}_{\boldsymbol{x}}, \mathcal{D}_{\boldsymbol{x}}), \quad (14)$$

where $\beta_2 > 0$ is a tunable hyperparameter. $\mathcal{L}_{\text{pred}}$ and $\mathcal{L}_{\text{recon}}$ are defined as $\mathcal{L}_{\text{pred}}(\mathcal{E}_{\boldsymbol{x}}, \mathcal{E}_{\boldsymbol{u}}, \mathcal{F}) = \frac{1}{n} \sum_{i=0}^{n-1} \left\| \mathcal{E}_{\boldsymbol{x}}\big(\boldsymbol{x}(t_{i+1})\big) - \mathcal{G}\Big(\mathcal{E}_{\boldsymbol{x}}\big(\boldsymbol{x}(t_i)\big), \mathcal{E}_{\boldsymbol{u}}\big(\boldsymbol{u}(t_i)\big)\Big) \right\|^2$ and $\mathcal{L}_{\text{recon}}(\mathcal{E}_{\boldsymbol{x}}, \mathcal{D}_{\boldsymbol{x}}) = \frac{1}{n} \sum_{i=1}^{n} \big\| \boldsymbol{x}(t_i) - \mathcal{D}_{\boldsymbol{x}} \circ \mathcal{E}_{\boldsymbol{x}}\big(\boldsymbol{x}(t_i)\big) \big\|^2$. Here, the operator $\circ$ denotes the composition of two functions. In experiments, $\mathcal{L}_{\text{recon}}$ also includes the reconstruction loss of the desired state where we want to stabilize the system. Figure 1a shows the overall framework for training DeepROM.

## 4.2 LEARNING CONTROL

Once we get a trained ROM of the form $\frac{d\boldsymbol{x}_{\text{R}}}{dt} = \mathcal{F}(\boldsymbol{x}_{\text{R}}, \boldsymbol{u}_{\text{R}})$ using the method proposed in section 4.1, the next goal is to design a controller for the system utilizing that ROM. Since our ROM is represented by DNNs, we need a data-driven method to develop the controller. We adopt the approach presented by Saha et al. (2021) for learning control laws for nonlinear systems, represented by DNNs. The core idea of the method is to hypothesize a target dynamics that is exponentially stable at the desired state and simultaneously learn a control policy to realize that target dynamics in the closed loop. A DNN is used to represent the vector field $\mathcal{F}_s : \mathbb{R}^{r_{\boldsymbol{x}}} \to \mathbb{R}^{r_{\boldsymbol{x}}}$ of the target dynamics $\frac{d\boldsymbol{x}_{\text{R}}}{dt} = \mathcal{F}_s(\boldsymbol{x}_{\text{R}})$. We use another DNN to represent a controller $\Pi : \mathbb{R}^{r_{\boldsymbol{x}}} \to \mathbb{R}^{d_{\boldsymbol{u}}}$ that provides the necessary actuation for a given reduced state $\boldsymbol{x}_{\text{R}}$: $\boldsymbol{u} = \Pi(\boldsymbol{x}_{\text{R}})$. This control $\boldsymbol{u}$ is then encoded by (trained) $\mathcal{E}_{\boldsymbol{u}}$ to its low-dimensional representation $\boldsymbol{u}_{\text{R}}$. Finally, the reduced state $\boldsymbol{x}_{\text{R}}$ and actuation $\boldsymbol{u}_{\text{R}}$ are fed to the (trained) ROM of $\frac{d\boldsymbol{x}_{\text{R}}}{dt} = \mathcal{F}(\boldsymbol{x}_{\text{R}}, \boldsymbol{u}_{\text{R}})$ to get $\mathcal{F}(\boldsymbol{x}_{\text{R}}, \boldsymbol{u}_{\text{R}})$. The overall framework for learning control is referred to as deep reduced-order control (DeepROC) and is shown in Figure 1b.

Our training objective is to minimize the difference between $\mathcal{F}(\boldsymbol{x}_{\text{R}}, \boldsymbol{u}_{\text{R}})$ and $\mathcal{F}_s(\boldsymbol{x}_{\text{R}})$, i.e.,

$$\mathcal{L}_{\text{ctrl}}(\mathcal{F}_s, \Pi) = \frac{1}{n} \sum_{i=1}^{n} \left\| \mathcal{F}\big(\mathcal{E}_{\boldsymbol{x}}(\boldsymbol{x}(t_i)), \mathcal{E}_{\boldsymbol{u}} \circ \Pi \circ \mathcal{E}_{\boldsymbol{x}}(\boldsymbol{x}(t_i))\big) - \mathcal{F}_s \circ \mathcal{E}_{\boldsymbol{x}}\big(\boldsymbol{x}(t_i)\big) \right\|^2. \quad (15)$$

To minimize the control effort, we add a regularization loss with (15), and the overall training objective for learning control is given by

$$\mathcal{L}_{\text{ctrl,reg}}(\mathcal{F}_s, \Pi) = \mathcal{L}_{\text{ctrl}}(\mathcal{F}_s, \Pi) + \beta_3 \, \frac{1}{n} \sum_{i=1}^{n} \left\| \Pi(\boldsymbol{x}_{\text{R}}(t_i)) \right\|^2, \tag{16}$$

where $\beta_3 > 0$ is a tunable hyperparameter. Here we jointly train the DNNs representing $\Pi$ and $\mathcal{F}_s$ only, whereas the previously-trained DNNs for $\mathcal{E}_{\boldsymbol{x}}, \mathcal{E}_{\boldsymbol{u}}$, and $\mathcal{F}$ are kept frozen. Once all the DNNs are trained, we only need $\mathcal{E}_{\boldsymbol{x}}$ and $\Pi$ during evaluation to generate actuation for the actual system, given a full-state observation: $\boldsymbol{u} = \Pi \circ \mathcal{E}_{\boldsymbol{x}}(\boldsymbol{x}) = \pi(\boldsymbol{x})$. As we mentioned earlier, we require the target dynamics, hypothesized by a DNN, to be exponentially stable at the desired state. Without loss of generality, we consider stability at $\boldsymbol{x}_{\text{R}} = \boldsymbol{0}$. The system can be stabilized at any desired state by adding a feedforward component to the control (see appendix A.1). Dynamics represented by a standard neural network is not stable at any equilibrium point, in general. Kolter & Manek (2019) showed that it is possible to design a DNN, by means of Lyapunov functions, to represent a dynamics that is exponentially stable at an equilibrium point. Accordingly, we represent our target dynamics as follows:

$$\frac{d\boldsymbol{x}_{\text{R}}}{dt} = \mathcal{F}_s(\boldsymbol{x}_{\text{R}}) = \mathcal{P}(\boldsymbol{x}_{\text{R}}) - \frac{\text{ReLU}\big(\nabla\mathcal{V}_{\text{R}}(\boldsymbol{x}_{\text{R}})^\top \mathcal{P}(\boldsymbol{x}_{\text{R}}) + \alpha\mathcal{V}_{\text{R}}(\boldsymbol{x}_{\text{R}})\big)}{\|\nabla\mathcal{V}_{\text{R}}(\boldsymbol{x}_{\text{R}})\|^2} \nabla\mathcal{V}_{\text{R}}(\boldsymbol{x}_{\text{R}}), \tag{17}$$

where $\alpha$ is a positive constant, $\text{ReLU}(z) = \max(0, z)$, $z \in \mathbb{R}$, and $\mathcal{V}_{\text{R}} : \mathbb{R}^{r_{\boldsymbol{x}}} \to \mathbb{R}$ is a candidate Lyapunov function. We use

$$\mathcal{V}_{\text{R}}(\boldsymbol{x}_{\text{R}}) = \boldsymbol{x}_{\text{R}}^\top \boldsymbol{K} \boldsymbol{x}_{\text{R}}, \tag{18}$$

where $\boldsymbol{K} \in \mathbb{R}^{r_{\boldsymbol{x}} \times r_{\boldsymbol{x}}}$ is a positive definite matrix. The target dynamics of (17) is exponentially stable at the origin, as shown in Kolter & Manek (2019).

## 5 EMPIRICAL RESULTS

### 5.1 BASELINES

The prediction performance of DeepROM is compared against DMDc and the Deep Koopman model (Morton et al. (2018)). The Deep Koopman model shares a similar DNN-based autoencoding structure as ours, with the distinction that its (reduced-order) dynamic model is linear. The method proposed by Morton et al. (2018) considers a model predictive scenario, where the state/system matrix of the linear reduced-order model is updated with online observations during operation while the input/control matrix is kept fixed. However, in contrast to the original method, we keep both matrices fixed during the control operation as we consider offline control design in this paper. For the same reason, we apply linear quadratic regulator (LQR) on the ROM obtained from the Deep Koopman method, instead of model predictive control, to compare the control performance with our method: DeepROC. The control performance is also compared against the reduced order controller obtained by applying LQR on the ROM derived from DMDc. Details on the neural network architectures and training settings for the Deep Koopman model are given in appendix F. The similarity between DMDc and LAROM can be visualized using the dynamic modes estimated in respective methods. Due to space limitations, these visualizations are provided in the appendix E.

### 5.2 REACTION–DIFFUSION SYSTEM STABILIZATION

For the first experiment, we consider the Newell-Whitehead-Segel reaction-diffusion equation which is used to describe various nonlinear physical systems including Rayleigh-Bénard convection. The considered system is a bistable system with $\pm 1$ as stable and $0$ as unstable equilibria. For the control task, we consider feedback stabilization of this system at the unstable equilibrium $0$, as studied by Kalise & Kunisch (2018). Details on the system definition, dataset generation, neural network architectures, and training settings are given in appendix C.

**Prediction performance.** First, we compare the performance of DeepROM, Deep Koopman model, and DMDc in the prediction task. Note, this example uses low-dimensional actuation (just a single variable). Accordingly, the control encoder $\mathcal{E}_{\boldsymbol{u}}$ is not used here. Figure 2(a) shows the quantitative comparison of the recursive multi-step predictions obtained using DMDc, Deep Koopman model, and DeepROM. The prediction error is computed as *normalized mean squared error* (NMSE) with respect to the solution obtained using the PDE solver. Prediction error increases more quickly for

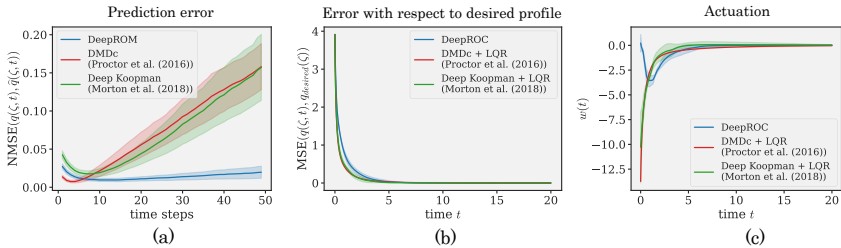

Figure 2: (a): Prediction performance of different methods in the reaction–diffusion example. The shaded interval shows the $95\%$ confidence interval around the mean from $100$ test sequences and $3$ different training instances. (b,c): Control performance of different methods in the reaction–diffusion example. The shaded interval shows the 1-standard deviation range around the mean from 3 different training instances.

DMDc and Deep Koopman than DeepROM as the linear ROMs become less accurate in the long term. A qualitative comparison of the prediction performance of the methods for an example sequence is given in the appendix C.5.

**Control performance.** Figures 2(b,c) show the control performance of DeepROC, Deep Koopman + LQR, and DMDc + LQR in the task of stabilizing the system at the unstable equilibrium 0 from an initial state $2 + \cos(2\pi\zeta)\cos(\pi\zeta)$. We use the following two metrics for comparison:

  (i)  mean squared error over time between the controlled solutions and the desired profile
 (ii)  the amount of actuation applied

All methods show similar closed-loop error profiles. However, DeepROC requires significantly less amount of actuation in comparison with DMDc + LQR and Deep Koopman + LQR to reach a similar steady-state error. DeepROC can account for the decaying nonlinear term $-q^3$ present in the system (47) and therefore learns to apply less actuation. A qualitative comparison of the uncontrolled solution and the controlled solutions obtained using the three methods is given in the appendix C.5.

For this example, we also compare performance with a model-based control method that optimizes the control input of a trained surrogate model through backpropagation. Specifically, we investigate the method proposed by Hwang et al. (2022). Such optimization technique is computationally expensive for time-dependent PDEs, particularly when a long trajectory is needed to be rolled out. We observed that we can optimize the control input only up to a certain time step. Though the system state reaches the target within this timeframe, it fails to stay stable since the target is an unstable equilibrium and the control input is no longer effective. We added this result in the supplementary (appendix C.5).

### 5.3 VORTEX SHEDDING SUPPRESSION IN FLUID

In this experiment, we consider modeling and suppressing vortex shedding in two-dimensional incompressible flow past a circular cylinder. This is a well-known problem (Schäfer et al. (1996)) and is of great importance for many engineering applications (Williamson (1996)). The density and kinematic viscosity of the fluid are chosen such that the Reynolds number is $Re = 50$, which is just above the cutoff for the onset of the vortex shedding (Williamson (1996)). In this case, vortices are created at the back of the cylinder and are shed periodically from the upper and lower surfaces of the cylinder forming a von Kármán vortex street (Morton et al. (2018)). Details on the problem setup, dataset generation, neural network architectures, and training settings are given in appendix D.

**Prediction performance.** Figure 3(a) shows the quantitative comparison of the recursive multi-step predictions, starting from $t = 0.1$, obtained using DMDc, Deep Koopman model, and DeepROM. The initial state is chosen at $t = 0.1$ because the fluid does not reach the observation region $\mathbb{W}$ before that time. The prediction error is computed as the *mean squared error* (MSE) with respect to the solution obtained using a PDE solver. DeepROM shows lower prediction error in comparison with DMDc. The Deep Koopman model shows better prediction performance than DeepROM and DMDc during the initial few steps. However, its accuracy deteriorates rapidly and eventually becomes comparable to that of DMDc. A qualitative comparison of the prediction performance of the three methods is given in the appendix D.5. Additionally, we compare the prediction performance of DeepROM for an unforced system with the transformer-based model VideoGPT (Yan et al. (2021)).

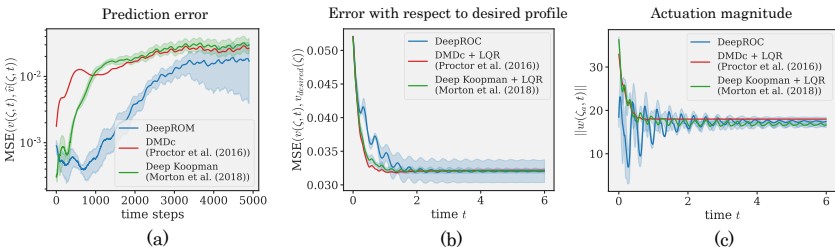

Figure 3: (a): Prediction performance of different methods in the fluid flow example. The shaded interval shows 1-standard deviation range around the mean from 3 training instances. (b, c): Control performance of different methods in the vortex shedding suppression task. The shaded interval shows 1-standard deviation range around the mean from 3 training instances.

Results of this experiment are provided in appendix D.5 and indicate that VideoGPT generates satisfactory predictions in the short term but falls short in capturing long-term dynamics.

**Control performance.** Figure 3(b,c) shows the control performance of DeepROC, Deep Koopman + LQR, and DMDc+LQR in the task of suppressing vortex shedding. The controllers of DeepROC and DMDc + LQR directly estimate the high-dimensional actuation distributed over space. However, the same technique proved ineffective in suppressing the shedding for Deep Koopman + LQR. Therefore, instead of directly estimating the distributed actuation, we utilize a low-dimensional representation of the actuation for Deep Koopman + LQR. We represent the distributed actuation as a linear combination of some space-dependent sinusoidal basis functions. The controller is designed to estimate the coefficients of those basis functions in the linear combination. Details are provided in appendix F.

We use the same metrics as the previous example for comparison except for actuation. Since distributed control is applied in this case, we use the magnitude of the actuation here. To reach a similar steady-state error, DeepROC takes a longer time compared to DMDc and Deep Koopman + LQR. DeepROM uses the least amount of actuation during the initial few steps, whereas Deep Koopman + LQR has the least steady-state actuation magnitude. A qualitative comparison of the uncontrolled solution and the controlled solutions obtained using the three methods is given in the appendix D.5.

**Comparison with full-order model-based control.** In assessing the benefits of employing a reduced-order model in contrast to a full-order model (FOM), we conduct an ablation study with an FOM and the identical control method applied directly to it. While FOM exhibits superior prediction accuracy for the initial steps, it experiences a rapid decline in accuracy over time. On the other hand, FOM + NI4C (Saha et al. (2021)) shows better performance than DeepROC for the control task. However, it is crucial to highlight that the advantage of employing a reduced-order model over a full-order model in control learning primarily resides in reduced computational complexity without a substantial compromise in accuracy. Effective FLOP count (#E-FLOPs) per training or prediction step for FOM is significantly higher (over 150X) than that of DeepROM despite a similar parameter count. The substantial computational disparity arises due to the NI4C's requirement for a continuous-time formulation of the model, necessitating numerical integration during both training and inference. Due to space limitations, the detailed quantitative comparisons of performance and computational costs are provided in the supplementary (appendix D.5).

## 6  CONCLUSION

We presented a framework for autoencoder-based modeling and control learning for PDE-driven dynamical systems. The proposed reduced-order modeling framework is grounded on the connection between dynamic mode decomposition for controlled systems and a linear autoencoding architecture that can be trained using gradient descent. As we showed in experiments, DeepROM offers better prediction accuracy than a linear ROM over a relatively longer prediction horizon when applied to nonlinear systems. However, this advantage does not always translate to significant improvement in control performance. Designing controllers for DNN-based models is a challenging task due to the standard difficulties associated with non-convex optimization. Nevertheless, we envision great prospects in solving many problems of control design for high-dimensional systems utilizing autoencoder-based models as they continue to demonstrate their effectiveness in the analysis and prediction of such systems.

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

# A ADDITIONAL PRELIMINARIES

## A.1 STABILIZATION OF CONTROLLED SYSTEMS

Suppose the function $f$ in (2) is locally Lipschitz and $(\boldsymbol{x} = \boldsymbol{0}, \boldsymbol{u} = \boldsymbol{0})$ is an equilibrium pair of the system, i.e., $f(\boldsymbol{0}, \boldsymbol{0}) = \boldsymbol{0}$. The system (2) is said to be *locally stabilizable* with respect to the equilibrium pair if there exists a locally Lipschitz function $\pi : \mathbb{X}_0 \to \mathbb{U}$, $\pi(\boldsymbol{0}) = \boldsymbol{0}$, defined on some neighborhood $\mathbb{X}_0 \subset \mathbb{X}$ of the origin $\boldsymbol{x} = \boldsymbol{0}$ for which the closed-loop system $\frac{d\boldsymbol{x}}{dt} = f(\boldsymbol{x}, \pi(\boldsymbol{x}))$ is locally *asymptotically stable*, i.e. $\|\boldsymbol{x}(t_0)\| < \delta$ implies $\lim_{t \to \infty} \boldsymbol{x}(t) = \boldsymbol{0}$ (Sontag (2013)).

Stability of the closed-loop system $\frac{d\boldsymbol{x}}{dt} = f(\boldsymbol{x}, \pi(\boldsymbol{x})) = h(\boldsymbol{x})$ at equilibrium points can be analyzed using the method of Lyapunov. Let $\mathcal{V} : \mathbb{X} \to \mathbb{R}$ be a continuously differentiable function such that

$$\mathcal{V}(\boldsymbol{0}) = 0, \quad \text{and} \quad \mathcal{V}(\boldsymbol{x}) > 0 \quad \forall \, \boldsymbol{x} \in \mathbb{X} \setminus \{\boldsymbol{0}\}, \tag{19}$$

and the time derivative of $\mathcal{V}$ along the trajectories

$$\frac{d\mathcal{V}}{dt} = \nabla \mathcal{V}(\boldsymbol{x})^\top \frac{d\boldsymbol{x}}{dt} = \nabla \mathcal{V}(\boldsymbol{x})^\top h(\boldsymbol{x}) \le 0 \quad \forall \, \boldsymbol{x} \in \mathbb{X}. \tag{20}$$

Then, the equilibrium point $\boldsymbol{x} = \boldsymbol{0}$ is *stable*, i.e., for each $\epsilon > 0$, there exists a $\delta = \delta(\epsilon) > 0$ such that $\|\boldsymbol{x}(t_0)\| < \delta$ implies $\|\boldsymbol{x}(t)\| < \epsilon$, $\forall t > t_0$. The function $\mathcal{V}$ with the above properties is called a Lyapunov function. If $\frac{d\mathcal{V}}{dt} < 0$ in some subset $\mathbb{X}_s \subset \mathbb{X} \setminus \{\boldsymbol{0}\}$, then $\boldsymbol{x} = \boldsymbol{0}$ is *locally asymptotically stable*. Moreover, if there exist positive constants $c_1, c_2, c_3$ and $c_4$ such that

$$c_1 \|\boldsymbol{x}\|^2 \le \mathcal{V}(\boldsymbol{x}) \le c_2 \|\boldsymbol{x}\|^2, \tag{21}$$

and

$$\nabla \mathcal{V}(\boldsymbol{x})^\top h(\boldsymbol{x}) \le -c_3 \|\boldsymbol{x}\|^2, \quad \forall \, \boldsymbol{x} \in \mathbb{X}_s, \tag{22}$$

then $\boldsymbol{x} = \boldsymbol{0}$ is *exponentially stable*, i.e., there exist positive constants $\delta, \lambda$ and $\gamma$ such that $\|\boldsymbol{x}(t)\| \le \lambda \|\boldsymbol{x}(t_0)\| e^{-\gamma(t - t_0)}$, $\forall \|\boldsymbol{x}(t_0)\| < \delta$ (Khalil (2002)).

Though the above formulation is for stabilization at the equilibrium point $\boldsymbol{x} = \boldsymbol{0}$, the same can be used for developing control to stabilize the system at any arbitrary point $\boldsymbol{x}_{ss}$. In that case, a steady-state control input $\boldsymbol{u}_{ss}$ is required that can maintain the equilibrium at $\boldsymbol{x}_{ss}$, i.e., $f(\boldsymbol{x}_{ss}, \boldsymbol{u}_{ss}) = \boldsymbol{0}$. The change of variables $\boldsymbol{x}_e = \boldsymbol{x} - \boldsymbol{x}_{ss}, \boldsymbol{u}_e = \boldsymbol{u} - \boldsymbol{u}_{ss}$ leads to a transformed system where we can apply the aforementioned formulation of stabilization. The overall control, in this case, $\boldsymbol{u} = \boldsymbol{u}_e + \boldsymbol{u}_{ss}$ comprises a feedback component $\boldsymbol{u}_e$ and a feedforward component $\boldsymbol{u}_{ss}$ (Khalil (2002)).

In this paper, we assume that the system we are aiming to stabilize at an equilibrium point is stabilizable in the sense of the aforementioned definition and criteria, i.e., there exists a continuously differentiable function $\mathcal{V}$ and a Lipschitz continuous control law $\pi$ such that criteria (19) and (20) are conformed.

## A.2 SOME PROPERTIES OF MATRICES USED FOR PROOFS

The proofs of the analytical results presented in this paper use the following properties of the rank (denoted by $\text{rank}(\cdot)$), the Kronecker product (denoted by $\otimes$) and vectorization of matrices (denoted by $\text{vec}(\cdot)$). All the definitions and properties are presented in the context of matrices over real numbers.

For any conformable matrices $\boldsymbol{D}$ and $\boldsymbol{E}$ such that $\boldsymbol{E}$ has full row-rank,

$$\text{rank}(\boldsymbol{D}\boldsymbol{E}) = \text{rank}(\boldsymbol{D}). \tag{23a}$$

For any real matrix $\boldsymbol{D}$,

$$\text{rank}(\boldsymbol{D}^\top \boldsymbol{D}) = \text{rank}(\boldsymbol{D}\boldsymbol{D}^\top) = \text{rank}(\boldsymbol{D}^\top) = \text{rank}(\boldsymbol{D}). \tag{23b}$$

For any matrices (of compatible dimensions) $\boldsymbol{D}, \boldsymbol{E}, \boldsymbol{F}$, and $\boldsymbol{H}$,

$$\text{vec}(\boldsymbol{D}\boldsymbol{E}\boldsymbol{F}^\top) = (\boldsymbol{F} \otimes \boldsymbol{D})\text{vec}(\boldsymbol{E}), \tag{24a}$$

$$(\boldsymbol{D} \otimes \boldsymbol{E})^\top = \boldsymbol{D}^\top \otimes \boldsymbol{E}^\top, \tag{24b}$$

$$(\boldsymbol{D} \otimes \boldsymbol{E})(\boldsymbol{F} \otimes \boldsymbol{H}) = (\boldsymbol{D}\boldsymbol{F} \otimes \boldsymbol{E}\boldsymbol{H}), \tag{24c}$$

whenever these quantities are defined. Furthermore, if $\boldsymbol{D}$ and $\boldsymbol{E}$ are symmetric and positive semidefinite (resp. positive definite), then $\boldsymbol{D} \otimes \boldsymbol{E}$ is symmetric and positive semidefinite (resp. positive definite), i.e.,

$$\boldsymbol{D} \succeq 0, \boldsymbol{E} \succeq 0 \implies (\boldsymbol{D} \otimes \boldsymbol{E}) \succeq 0; \quad \boldsymbol{D} \succ 0, \boldsymbol{E} \succ 0 \implies (\boldsymbol{D} \otimes \boldsymbol{E}) \succ 0. \tag{24d}$$

Proofs of (23) and (24) can be found in (Matsaglia & PH Styan (1974)) and (Magnus & Neudecker (1986)), respectively.

To derive the results presented in corollary (4.1.1.1), we use the following definitions of the Moore-Penrose inverse of a matrix (denoted by $(\cdot)^+$). For any matrix $\boldsymbol{D}$ and its (full) SVD, i.e., $\boldsymbol{D} = \boldsymbol{U}_D \boldsymbol{\Sigma}_D \boldsymbol{V}_D^\top$,

$$\boldsymbol{D}^+ = (\boldsymbol{D}^\top \boldsymbol{D})^{-1} \boldsymbol{D}^\top, \quad \text{when } (\boldsymbol{D}^\top \boldsymbol{D})^{-1} \text{ exists}, \tag{25a}$$

$$\boldsymbol{D}^+ = \boldsymbol{D}^\top (\boldsymbol{D} \boldsymbol{D}^\top)^{-1}, \quad \text{when } (\boldsymbol{D} \boldsymbol{D}^\top)^{-1} \text{ exists}, \tag{25b}$$

$$\boldsymbol{D}^+ = \boldsymbol{V}_D \boldsymbol{\Sigma}_D^+ \boldsymbol{U}_D^\top, \tag{25c}$$

$$\boldsymbol{D}^+ = \lim_{\varepsilon \to 0} (\boldsymbol{D}^\top \boldsymbol{D} + \varepsilon^2 \boldsymbol{I})^{-1} \boldsymbol{D}^\top = \lim_{\varepsilon \to 0} \boldsymbol{D}^\top (\boldsymbol{D} \boldsymbol{D}^\top + \varepsilon^2 \boldsymbol{I})^{-1}, \tag{25d}$$

where $\boldsymbol{I}$ is the identity matrix of compatible dimension. The proof of (25d) can be found in (Albert (1972)).

## B PROOFS

This section details the proofs for the results presented in section 4. To prove Theorem 4.1.1, we use some well-known results, summarized as the following lemma in (Baldi & Hornik (1989)), for linear least-squares optimization.

**Lemma B.0.1.** *The quadratic function $L(\boldsymbol{z}) = \|\boldsymbol{y} - \boldsymbol{M}\boldsymbol{z}\|^2 = \boldsymbol{y}^\top \boldsymbol{y} - 2\boldsymbol{y}^\top \boldsymbol{M}\boldsymbol{z} + \boldsymbol{z}^\top \boldsymbol{M}^\top \boldsymbol{M}\boldsymbol{z}$ is convex, and a point $\boldsymbol{z}$ globally minimizes $L$ if and only if $\nabla L(\boldsymbol{z}) = 0$, or equivalently, $\boldsymbol{M}^\top \boldsymbol{M}\boldsymbol{z} = \boldsymbol{M}^\top \boldsymbol{y}$. Furthermore, if $\boldsymbol{M}^\top \boldsymbol{M} \succ 0$, i.e., positive definite, then $L$ is strictly convex and reaches its unique minimum for $\boldsymbol{z} = (\boldsymbol{M}^\top \boldsymbol{M})^{-1} \boldsymbol{M}^\top \boldsymbol{y}$.*

### B.1 PROOF OF THEOREM 4.1.1

**Theorem 4.1.1.** *Consider the following objective function*

$$L_{\text{pred}}(\boldsymbol{E}_{\boldsymbol{x}}, \boldsymbol{G}) = \frac{1}{n} \sum_{i=0}^{n-1} \left\| \boldsymbol{E}_{\boldsymbol{x}} \boldsymbol{x}(t_{i+1}) - \boldsymbol{G} \boldsymbol{E}_{\boldsymbol{x}\boldsymbol{u}} \boldsymbol{\omega}(t_i) \right\|^2, \tag{8}$$

*where $\boldsymbol{G} = [\boldsymbol{A}_{\text{R}} \ \boldsymbol{B}_{\text{R}}] \in \mathbb{R}^{r_{\boldsymbol{x}} \times (r_{\boldsymbol{x}} + d_{\boldsymbol{u}})}, \boldsymbol{E}_{\boldsymbol{x}\boldsymbol{u}} = \begin{bmatrix} \boldsymbol{E}_{\boldsymbol{x}} & \boldsymbol{0} \\ \boldsymbol{0} & \boldsymbol{I}_{d_{\boldsymbol{u}}} \end{bmatrix} \in \mathbb{R}^{(r_{\boldsymbol{x}} + d_{\boldsymbol{u}}) \times (d_{\boldsymbol{x}} + d_{\boldsymbol{u}})}, \boldsymbol{I}_{d_{\boldsymbol{u}}}$ being the identity matrix of order $d_{\boldsymbol{u}}$. For any fixed matrix $\boldsymbol{E}_{\boldsymbol{x}}$, the objective function $L_{\text{pred}}$ is convex in the coefficients of $\boldsymbol{G}$ and attains its minimum for any $\boldsymbol{G}$ satisfying*

$$\boldsymbol{G} \boldsymbol{E}_{\boldsymbol{x}\boldsymbol{u}} \boldsymbol{\Omega} \boldsymbol{\Omega}^\top \boldsymbol{E}_{\boldsymbol{x}\boldsymbol{u}}^\top = \boldsymbol{E}_{\boldsymbol{x}} \boldsymbol{Y} \boldsymbol{\Omega}^\top \boldsymbol{E}_{\boldsymbol{x}\boldsymbol{u}}^\top, \tag{9}$$

*where $\boldsymbol{Y}$ and $\boldsymbol{\Omega}$ are the data matrices as defined in section (3.2). If $\boldsymbol{E}_{\boldsymbol{x}}$ has full rank $r_{\boldsymbol{x}}$, and $\boldsymbol{\Omega} \boldsymbol{\Omega}^\top$ is non-singular, then $L_{\text{pred}}$ is strictly convex and has a unique minimum for*

$$\boldsymbol{G} = [\boldsymbol{A}_{\text{R}} \ \boldsymbol{B}_{\text{R}}] = \boldsymbol{E}_{\boldsymbol{x}} \boldsymbol{Y} \boldsymbol{\Omega}^\top \boldsymbol{E}_{\boldsymbol{x}\boldsymbol{u}}^\top (\boldsymbol{E}_{\boldsymbol{x}\boldsymbol{u}} \boldsymbol{\Omega} \boldsymbol{\Omega}^\top \boldsymbol{E}_{\boldsymbol{x}\boldsymbol{u}}^\top)^{-1}. \tag{10}$$

*Proof.* We can write $L_{\text{pred}}(\boldsymbol{E}_{\boldsymbol{x}}, \boldsymbol{G})$ as follows,

$$\begin{aligned} L_{\text{pred}}(\boldsymbol{E}_{\boldsymbol{x}}, \boldsymbol{G}) &= \frac{1}{n} \sum_{i=0}^{n-1} \left\| \boldsymbol{E}_{\boldsymbol{x}} \boldsymbol{x}(t_{i+1}) - \boldsymbol{G} \boldsymbol{E}_{\boldsymbol{x}\boldsymbol{u}} \boldsymbol{\omega}(t_i) \right\|^2 \\ &= \left\| \text{vec}(\boldsymbol{E}_{\boldsymbol{x}} \boldsymbol{Y}) - \text{vec}(\boldsymbol{G} \boldsymbol{E}_{\boldsymbol{x}\boldsymbol{u}} \boldsymbol{\Omega}) \right\|^2 \\ &= \left\| \text{vec}(\boldsymbol{E}_{\boldsymbol{x}} \boldsymbol{Y}) - (\boldsymbol{\Omega}^\top \boldsymbol{E}_{\boldsymbol{x}\boldsymbol{u}}^\top \otimes \boldsymbol{I}_{r_{\boldsymbol{x}}}) \text{vec}(\boldsymbol{G}) \right\|^2. \end{aligned} \tag{26}$$

The third equality is obtained using (24a). For fixed $\boldsymbol{E_x}$, we can apply Lemma B.0.1 to (26): (26) is convex in coefficient of $\boldsymbol{G}$, and $\boldsymbol{G}$ corresponds to a global minimum of $L_{\text{pred}}$ if and only if

$$(\boldsymbol{\Omega}^\top \boldsymbol{E}_{\boldsymbol{xu}}^\top \otimes \boldsymbol{I}_{r_x})^\top (\boldsymbol{\Omega}^\top \boldsymbol{E}_{\boldsymbol{xu}}^\top \otimes \boldsymbol{I}_{r_x}) \text{vec}(\boldsymbol{G}) = (\boldsymbol{\Omega}^\top \boldsymbol{E}_{\boldsymbol{xu}}^\top \otimes \boldsymbol{I}_{r_x})^\top \text{vec}(\boldsymbol{E_x Y}). \qquad (27)$$

Using (24b) and (24c), we can write (27) as

$$(\boldsymbol{E}_{\boldsymbol{xu}} \boldsymbol{\Omega} \boldsymbol{\Omega}^\top \boldsymbol{E}_{\boldsymbol{xu}}^\top \otimes \boldsymbol{I}_{r_x}) \text{vec}(\boldsymbol{G}) = (\boldsymbol{E}_{\boldsymbol{xu}} \boldsymbol{\Omega} \otimes \boldsymbol{I}_{r_x}) \text{vec}(\boldsymbol{E_x Y}). \qquad (28)$$

Applying (24a) on (28), we get $\boldsymbol{G E}_{\boldsymbol{xu}} \boldsymbol{\Omega} \boldsymbol{\Omega}^\top \boldsymbol{E}_{\boldsymbol{xu}}^\top = \boldsymbol{E_x Y \Omega}^\top \boldsymbol{E}_{\boldsymbol{xu}}^\top$, i.e., (9).

If $\boldsymbol{E_x}$ has full rank $r_x$, then $\boldsymbol{E}_{\boldsymbol{xu}} = \begin{bmatrix} \boldsymbol{E_x} & \boldsymbol{0} \\ \boldsymbol{0} & \boldsymbol{I}_{d_u} \end{bmatrix} \in \mathbb{R}^{(r_x+d_u)\times(d_x+d_u)}$ has full rank $(r_x + d_u)$. If $\boldsymbol{\Omega}\boldsymbol{\Omega}^\top \in \mathbb{R}^{(d_x+d_u)\times(d_x+d_u)}$ is non-singular, then $\boldsymbol{\Omega}$ has full row-rank $(d_x + d_u)$. Consequently, using (23a) and (23b), we have

$$\text{rank}(\boldsymbol{E}_{\boldsymbol{xu}} \boldsymbol{\Omega} \boldsymbol{\Omega}^\top \boldsymbol{E}_{\boldsymbol{xu}}^\top) = \text{rank}(\boldsymbol{E}_{\boldsymbol{xu}} \boldsymbol{\Omega}) = \text{rank}(\boldsymbol{E}_{\boldsymbol{xu}}) = r_x + d_u. \qquad (29)$$

Hence the symmetric positive semidefinite matrix $\boldsymbol{E}_{\boldsymbol{xu}} \boldsymbol{\Omega} \boldsymbol{\Omega}^\top \boldsymbol{E}_{\boldsymbol{xu}}^\top$ has full rank and therefore positive definite. Using (24b), (24c), and (24d), we can see that $(\boldsymbol{\Omega}^\top \boldsymbol{E}_{\boldsymbol{xu}}^\top \otimes \boldsymbol{I}_{r_x})^\top (\boldsymbol{\Omega}^\top \boldsymbol{E}_{\boldsymbol{xu}}^\top \otimes \boldsymbol{I}_{r_x}) = (\boldsymbol{E}_{\boldsymbol{xu}} \boldsymbol{\Omega} \boldsymbol{\Omega}^\top \boldsymbol{E}_{\boldsymbol{xu}}^\top \otimes \boldsymbol{I}_{r_x})$ is positive definite as well. Therefore, by Lemma B.0.1, (26) is strictly convex in the coefficients of $\boldsymbol{G}$ and has a unique minimum. Since $\boldsymbol{E}_{\boldsymbol{xu}} \boldsymbol{\Omega} \boldsymbol{\Omega}^\top \boldsymbol{E}_{\boldsymbol{xu}}^\top \succ 0$, it is invertible. Hence, from (9), we can say that the unique minimum of (26) is reached at $\boldsymbol{G} = \boldsymbol{E_x Y \Omega}^\top \boldsymbol{E}_{\boldsymbol{xu}}^\top (\boldsymbol{E}_{\boldsymbol{xu}} \boldsymbol{\Omega} \boldsymbol{\Omega}^\top \boldsymbol{E}_{\boldsymbol{xu}}^\top)^{-1}$, i.e., (10). ∎

## B.2 AN ALTERNATIVE REPRESENTATION OF (10)

Here we provide a possible alternative representation of (10) required to prove corollary 4.1.1.1.

**Lemma B.2.1.** *Consider the (full) SVD of the data matrix $\boldsymbol{\Omega}$ given by $\boldsymbol{\Omega} = \boldsymbol{U_\Omega} \boldsymbol{\Sigma_\Omega} \boldsymbol{V_\Omega}^\top$, where $\boldsymbol{U_\Omega} \in \mathbb{R}^{(d_x+d_u)\times(d_x+d_u)}$, $\boldsymbol{\Sigma_\Omega} \in \mathbb{R}^{(d_x+d_u)\times n}$, and $\boldsymbol{V_\Omega} \in \mathbb{R}^{n\times n}$. (10) can be expressed as*

$$\boldsymbol{G} = \lim_{\varepsilon \to 0} \boldsymbol{E_x Y V_\Omega} (\boldsymbol{\Sigma_\Omega}^\top \boldsymbol{U_\Omega}^\top \boldsymbol{E}_{\boldsymbol{xu}}^\top \boldsymbol{E}_{\boldsymbol{xu}} \boldsymbol{U_\Omega} \boldsymbol{\Sigma_\Omega} + \varepsilon^2 \boldsymbol{I}_n)^{-1} \boldsymbol{\Sigma_\Omega}^\top \boldsymbol{U_\Omega}^\top \boldsymbol{E}_{\boldsymbol{xu}}^\top. \qquad (30)$$

*Proof.* Replacing $\boldsymbol{\Omega}$ with its SVD in (10) we get,

$$\begin{aligned}
\boldsymbol{G} &= \boldsymbol{E_x Y V_\Omega} \boldsymbol{\Sigma_\Omega}^\top \boldsymbol{U_\Omega}^\top \boldsymbol{E}_{\boldsymbol{xu}}^\top (\boldsymbol{E}_{\boldsymbol{xu}} \boldsymbol{U_\Omega} \boldsymbol{\Sigma_\Omega} \boldsymbol{V_\Omega}^\top \boldsymbol{V_\Omega} \boldsymbol{\Sigma_\Omega}^\top \boldsymbol{U_\Omega}^\top \boldsymbol{E}_{\boldsymbol{xu}}^\top)^{-1} \\
&= \boldsymbol{E_x Y V_\Omega} \boldsymbol{\Sigma_\Omega}^\top \boldsymbol{U_\Omega}^\top \boldsymbol{E}_{\boldsymbol{xu}}^\top (\boldsymbol{E}_{\boldsymbol{xu}} \boldsymbol{U_\Omega} \boldsymbol{\Sigma_\Omega} \boldsymbol{\Sigma_\Omega}^\top \boldsymbol{U_\Omega}^\top \boldsymbol{E}_{\boldsymbol{xu}}^\top)^{-1} \\
&= \boldsymbol{E_x Y V_\Omega} (\boldsymbol{E}_{\boldsymbol{xu}} \boldsymbol{U_\Omega} \boldsymbol{\Sigma_\Omega})^+ \qquad (31)
\end{aligned}$$

The second equality is due to the orthogonality of $\boldsymbol{V_\Omega}$. The third equality is obtained using (25b). Substituting $(\boldsymbol{E}_{\boldsymbol{xu}} \boldsymbol{U_\Omega} \boldsymbol{\Sigma_\Omega})^+$ with the *limit definition* (25d) of the Moore-Penrose inverse, we get

$$\boldsymbol{G} = \lim_{\varepsilon \to 0} \boldsymbol{E_x Y V_\Omega} (\boldsymbol{\Sigma_\Omega}^\top \boldsymbol{U_\Omega}^\top \boldsymbol{E}_{\boldsymbol{xu}}^\top \boldsymbol{E}_{\boldsymbol{xu}} \boldsymbol{U_\Omega} \boldsymbol{\Sigma_\Omega} + \varepsilon^2 \boldsymbol{I}_n)^{-1} \boldsymbol{\Sigma_\Omega}^\top \boldsymbol{U_\Omega}^\top \boldsymbol{E}_{\boldsymbol{xu}}^\top. \qquad (32)$$

∎

## B.3 PROOF OF COROLLARY 4.1.1.1

**Corollary 4.1.1.1.** *Consider the (full) SVD of the data matrix $\boldsymbol{\Omega}$ given by $\boldsymbol{\Omega} = \boldsymbol{U_\Omega} \boldsymbol{\Sigma_\Omega} \boldsymbol{V_\Omega}^\top$, where $\boldsymbol{U_\Omega} \in \mathbb{R}^{(d_x+d_u)\times(d_x+d_u)}$, $\boldsymbol{\Sigma_\Omega} \in \mathbb{R}^{(d_x+d_u)\times n}$, and $\boldsymbol{V_\Omega} \in \mathbb{R}^{n\times n}$. If $\boldsymbol{E_x} = \widehat{\boldsymbol{U}}_{\boldsymbol{Y}}^\top$ and $\boldsymbol{\Omega}\boldsymbol{\Omega}^\top$ is non-singular, then the solution for $\boldsymbol{G} = [\boldsymbol{A}_{\text{R}} \ \ \boldsymbol{B}_{\text{R}}]$ corresponding to the unique minimum of $L_{pred}$ can be expressed as*

$$\boldsymbol{A}_R = \widehat{\boldsymbol{U}}_{\boldsymbol{Y}}^\top \boldsymbol{Y V_\Omega} \Sigma^* \boldsymbol{U}_{\boldsymbol{\Omega},1}^\top \widehat{\boldsymbol{U}}_{\boldsymbol{Y}}, \quad \text{and} \quad \boldsymbol{B}_R = \widehat{\boldsymbol{U}}_{\boldsymbol{Y}}^\top \boldsymbol{Y V_\Omega} \Sigma^* \boldsymbol{U}_{\boldsymbol{\Omega},2}^\top, \qquad (11)$$

*where $[\boldsymbol{U}_{\boldsymbol{\Omega},1}^\top \ \ \boldsymbol{U}_{\boldsymbol{\Omega},2}^\top] = \boldsymbol{U_\Omega}^\top$ with $\boldsymbol{U}_{\boldsymbol{\Omega},1} \in \mathbb{R}^{d_x \times (d_x+d_u)}, \boldsymbol{U}_{\boldsymbol{\Omega},2} \in \mathbb{R}^{d_u \times (d_x+d_u)}$, and $\Sigma^* = \lim_{\varepsilon \to 0} (\boldsymbol{\Sigma_\Omega}^\top \boldsymbol{U}_{\boldsymbol{\Omega},1}^\top \widehat{\boldsymbol{U}}_{\boldsymbol{Y}} \widehat{\boldsymbol{U}}_{\boldsymbol{Y}}^\top \boldsymbol{U}_{\boldsymbol{\Omega},1} \boldsymbol{\Sigma_\Omega} + \boldsymbol{\Sigma_\Omega}^\top \boldsymbol{U}_{\boldsymbol{\Omega},2}^\top \boldsymbol{U}_{\boldsymbol{\Omega},2} \boldsymbol{\Sigma_\Omega} + \varepsilon^2 \boldsymbol{I}_n)^{-1} \boldsymbol{\Sigma_\Omega}^\top$.*

*Proof.* By the definition of truncated SVD, the columns of $\widehat{U}_Y$ are orthonormal. Therefore, $\widehat{U}_Y^\top$ has full row-rank $r_x$. Hence, by theorem 4.1.1 and lemma B.2.1, if $E_x = \widehat{U}_Y^\top$, and $\Omega\Omega^\top$ is non-singular, then the unique minimum of $L_{\text{pred}}$, is reached when

$$G = \widehat{U}_Y^\top Y V_\Omega (E_{xu} U_\Omega \Sigma_\Omega)^+ = \lim_{\varepsilon \to 0} \widehat{U}_Y^\top Y V_\Omega (\Sigma_\Omega^\top U_\Omega^\top E_{xu}^\top E_{xu} U_\Omega \Sigma_\Omega + \varepsilon^2 I_n)^{-1} \Sigma_\Omega^\top U_\Omega^\top E_{xu}^\top. \tag{33}$$

Now, substituting $E_x = \widehat{U}_Y^\top$ in $E_{xu}$, and using the partition $U_\Omega^\top = [U_{\Omega,1}^\top \ U_{\Omega,2}^\top]$, where $U_{\Omega,1} \in \mathbb{R}^{d_x \times (d_x + d_u)}$, $U_{\Omega,2} \in \mathbb{R}^{d_u \times (d_x + d_u)}$, we get

$$E_{xu} U_\Omega = \begin{bmatrix} \widehat{U}_Y^\top & 0 \\ 0 & I_{d_u} \end{bmatrix} \begin{bmatrix} U_{\Omega,1} \\ U_{\Omega,2} \end{bmatrix} = \begin{bmatrix} \widehat{U}_Y^\top U_{\Omega,1} \\ U_{\Omega,2} \end{bmatrix}, \tag{34}$$

and

$$U_\Omega^\top E_{xu}^\top E_{xu} U_\Omega = \begin{bmatrix} U_{\Omega,1}^\top \widehat{U}_Y & U_{\Omega,2}^\top \end{bmatrix} \begin{bmatrix} \widehat{U}_Y^\top U_{\Omega,1} \\ U_{\Omega,2} \end{bmatrix} = U_{\Omega,1}^\top \widehat{U}_Y \widehat{U}_Y^\top U_{\Omega,1} + U_{\Omega,2}^\top U_{\Omega,2}. \tag{35}$$

Plugging (34) and (35) into (33) leads to

$$G = \lim_{\varepsilon \to 0} \widehat{U}_Y^\top Y V_\Omega (\Sigma_\Omega^\top U_{\Omega,1}^\top \widehat{U}_Y \widehat{U}_Y^\top U_{\Omega,1} \Sigma_\Omega + \Sigma_\Omega^\top U_{\Omega,2}^\top U_{\Omega,2} \Sigma_\Omega + \varepsilon^2 I_n)^{-1} \Sigma_\Omega^\top \begin{bmatrix} U_{\Omega,1}^\top \widehat{U}_Y & U_{\Omega,2}^\top \end{bmatrix}. \tag{36}$$

Defining $\Sigma^* \stackrel{\triangle}{=} \lim_{\varepsilon \to 0}(\Sigma_\Omega^\top U_{\Omega,1}^\top \widehat{U}_Y \widehat{U}_Y^\top U_{\Omega,1} \Sigma_\Omega + \Sigma_\Omega^\top U_{\Omega,2}^\top U_{\Omega,2} \Sigma_\Omega + \varepsilon^2 I_n)^{-1} \Sigma_\Omega^\top$, we can split (36) into

$$A_{\text{R}} = \widehat{U}_Y^\top Y V_\Omega \Sigma^* U_{\Omega,1}^\top \widehat{U}_Y, \quad \text{and} \quad B_{\text{R}} = \widehat{U}_Y^\top Y V_\Omega \Sigma^* U_{\Omega,2}^\top,$$

which is (11). ∎

## B.4 THE CASE WHEN $\Omega\Omega^\top$ NOT INVERTIBLE

When the covariance matrix $\Omega\Omega^\top$ is not invertible, which is always true if $n < d_x + d_u$, the matrix $E_{xu}\Omega\Omega^\top E_{xu}^\top$ is not guaranteed to be invertible. In that case, the minimum of $L_{\text{pred}}$ corresponds to infinitely many solutions for $G$. However, minimizing $L_{\text{pred}}$ with added $\ell_2$ regularization, i.e., $L_{\text{pred,reg}}(E_x, G) = L_{\text{pred}}(E_x, G) + \beta\|\text{vec}(G)\|^2$ provides a unique solution for $G$, for a fixed $E_x$. We have the following result.

**Theorem B.4.1.** *For any fixed matrix $E_x$ and $\beta > 0$, the objective function $L_{\text{pred,reg}}(E_x, G) = L_{\text{pred}}(E_x, G) + \beta\|vec(G)\|^2$ is strictly convex in the coefficients of $G$, and the global minimum of $L_{\text{pred,reg}}$ corresponds to the unique solution for $G$, given by*

$$G = E_x Y \Omega^\top E_{xu}^\top (E_{xu} \Omega\Omega^\top E_{xu}^\top + \beta I_{r_x + d_u})^{-1}. \tag{37}$$

*Proof.* $L_{\text{pred,reg}}(E_x, G)$ can be written as, using (24a-c),

$$\begin{aligned} L_{\text{pred,reg}}(E_x, G) &= \left\|\text{vec}(E_x Y) - (\Omega^\top E_{xu}^\top \otimes I_{r_x})\text{vec}(G)\right\|^2 + \beta\|\text{vec}(G)\|^2 \\ &= \text{vec}(E_x Y)^\top \text{vec}(E_x Y) - 2\text{vec}(E_x Y)^\top (\Omega^\top E_{xu}^\top \otimes I_{r_x})\text{vec}(G) \\ &\quad + \text{vec}(G)^\top (E_{xu} \Omega\Omega^\top E_{xu}^\top \otimes I_{r_x} + \beta I_{r_x(r_x + d_u)})\text{vec}(G) \end{aligned}$$

$E_{xu}\Omega\Omega^\top E_{xu}^\top$ is a symmetric positive semidefinite matrix, irrespective of whether it has full rank or not. Hence, by (24d), $E_{xu}\Omega\Omega^\top E_{xu}^\top \otimes I_{r_x}$ is symmetric positive semidefinite. Consequently, for any $\beta > 0$, $E_{xu}\Omega\Omega^\top E_{xu}^\top \otimes I_{r_x} + \beta I_{r_x(r_x + d_u)}$ is positive definite. According to lemma B.0.1, $L_{\text{pred,reg}}$ is therefore strictly convex in the coefficients of $G$ and globally minimized when $\nabla L_{\text{pred,reg}} = 0$. The unique solution of (37) can be derived in the same manner as theorem 4.1.1. ∎

**Remark.** Replacing $\Omega$ with its SVD in (37) we get,

$$G = E_x Y V_\Omega \Sigma_\Omega^\top U_\Omega^\top E_{xu}^\top (E_{xu} U_\Omega \Sigma_\Omega \Sigma_\Omega^\top U_\Omega^\top E_{xu}^\top + \beta I_{r_x + d_u})^{-1}. \tag{38}$$

In the limit $\beta \to 0^+$, (38) converges to (31).

## B.5 DMDc THROUGH A LINEAR AUTOENCODING STRUCTURE

Here we present a linear autoencoding structure that leads to a linear ROM exactly resembling the DMDc solution when $E_x = \widehat{U}_Y^\top$. However, its DNN-based nonlinear counterpart does not actually offer dimensionality reduction.

**Theorem B.5.1.** *Consider the following objective function*

$$L_{\text{pred,alt}}(E_x, \widetilde{G}) = \frac{1}{n}\sum_{i=0}^{n-1}\left\|E_x x(t_{i+1}) - \widetilde{G}\omega(t_i)\right\|^2, \tag{39}$$

*where $\widetilde{G} \in \mathbb{R}^{r_x \times (d_x + d_u)}$. For any fixed matrix $E_x$, the objective function $L_{\text{pred,alt}}$ is convex in the coefficients of $\widetilde{G}$ and attains its minimum for any $\widetilde{G}$ satisfying*

$$\widetilde{G}\Omega\Omega^\top = E_x Y \Omega^\top, \tag{40}$$

*where $Y$ and $\Omega$ are the data matrices as defined in section (3.2). If $\Omega\Omega^\top$ is non-singular, then $L_{\text{pred,alt}}$ is strictly convex and has a unique minimum for*

$$\widetilde{G} = E_x Y \Omega^\top (\Omega\Omega^\top)^{-1}. \tag{41}$$

*Proof.* The proof is very similar to the proof of theorem 4.1.1. Using (24a), we can write $L_{\text{pred,alt}}(E_x, \widetilde{G})$ as follows,

$$\begin{aligned}
L_{\text{pred,alt}}(E_x, \widetilde{G}) &= \frac{1}{n}\sum_{i=0}^{n-1}\left\|E_x x(t_{i+1}) - \widetilde{G}\omega(t_i)\right\|^2 \\
&= \left\|\text{vec}(E_x Y) - \text{vec}(\widetilde{G}\Omega)\right\|^2 \\
&= \left\|\text{vec}(E_x Y) - (\Omega^\top \otimes I_{r_x})\text{vec}(\widetilde{G})\right\|^2.
\end{aligned} \tag{42}$$

For fixed $E_x$, applying Lemma B.0.1 to (42), we can say $L_{\text{pred,alt}}$ is convex in the coefficients of $\widetilde{G}$, and $\widetilde{G}$ corresponds to a global minimum of $L_{\text{pred,alt}}$ if and only if

$$(\Omega^\top \otimes I_{r_x})^\top(\Omega^\top \otimes I_{r_x})\text{vec}(\widetilde{G}) = (\Omega^\top \otimes I_{r_x})^\top\text{vec}(E_x Y). \tag{43}$$

Using (24a-c), we can write (43) as $\widetilde{G}\Omega\Omega^\top = E_x Y \Omega^\top$, which is (40).

If $\Omega\Omega^\top$ is non-singular, then it is symmetric positive definite. Using (24b-d), we can see that $(\Omega^\top \otimes I_{r_x})^\top(\Omega^\top \otimes I_{r_x}) = (\Omega\Omega^\top \otimes I_{r_x})$ is positive definite as well. Therefore, by Lemma B.0.1, (42) is strictly convex in coefficient in $\widetilde{G}$ and has a unique minimum. In that case, from (40), we can say that the unique minimum of (42) is reached at $\widetilde{G} = E_x Y \Omega^\top (\Omega\Omega^\top)^{-1}$, i.e., (41). ∎

**Corollary B.5.1.1.** *Consider the (full) SVD of the data matrix $\Omega$ given by $\Omega = U_\Omega \Sigma_\Omega V_\Omega^\top$, where $U_\Omega \in \mathbb{R}^{(d_x + d_u) \times (d_x + d_u)}$, $\Sigma_\Omega \in \mathbb{R}^{(d_x + d_u) \times n}$, and $V_\Omega \in \mathbb{R}^{n \times n}$. If $E_x = \widehat{U}_Y^\top$ and $\Omega\Omega^\top$ is non-singular, then the solution for $\widetilde{G}$ corresponding to the unique minimum of $L_{\text{pred,alt}}$ can be expressed as*

$$\widetilde{G} = \widehat{U}_Y^\top Y V_\Omega \Sigma_\Omega^+ U_\Omega^\top. \tag{44}$$

*Proof.* By theorem B.5.1, if $E_x = \widehat{U}_Y^\top$, and $\Omega\Omega^\top$ is non-singular, then the unique minimum of $L_{\text{pred,alt}}$ is reached when

$$\widetilde{G} = \widehat{U}_Y^\top Y \Omega^\top (\Omega\Omega^\top)^{-1} = \widehat{U}_Y^\top Y \Omega^+ \tag{45}$$

The second equality is due to (25b). Substituting $\Omega^+$ with its *SVD definition* (25c) into (45), we get $\widehat{U}_Y^\top Y V_\Omega \Sigma_\Omega^+ U_\Omega^\top$, which is (44). ∎

**Remark.** From (39), it can be seen that $\widetilde{G}$ maps the concatenated vector, $\boldsymbol{\omega}(t_i)$, of full state and actuation to the next reduce state $\boldsymbol{x}_{\mathrm{R}}(t_{i+1})$. We can partition (44) as $\widetilde{G} = \widehat{\boldsymbol{U}}_{\boldsymbol{Y}}^{\top} \boldsymbol{Y} \boldsymbol{V}_{\boldsymbol{\Omega}} \Sigma_{\boldsymbol{\Omega}}^{+} [\boldsymbol{U}_{\boldsymbol{\Omega},1}^{\top} \ \boldsymbol{U}_{\boldsymbol{\Omega},2}^{\top}] = [\widetilde{\boldsymbol{A}} \ \widetilde{\boldsymbol{B}}]$ to separate out the blocks corresponding to state and actuation. Here, $\boldsymbol{U}_{\boldsymbol{\Omega},1}, \boldsymbol{U}_{\boldsymbol{\Omega},2}$ are the same as defined in corollary 4.1.1.1, and $\widetilde{\boldsymbol{A}} \in \mathbb{R}^{r_{\boldsymbol{x}} \times d_{\boldsymbol{x}}}, \widetilde{\boldsymbol{B}} \in \mathbb{R}^{r_{\boldsymbol{x}} \times d_{\boldsymbol{u}}}$. Now, if we post-multiply $\widetilde{\boldsymbol{A}}$ with $\boldsymbol{E}_{\boldsymbol{x}}^{\top} = \widehat{\boldsymbol{U}}_{\boldsymbol{Y}} \in \mathbb{R}^{d_{\boldsymbol{x}} \times r_{\boldsymbol{x}}}$, we get a ROM

$$\widetilde{\boldsymbol{A}}_{\mathrm{R}} = \widetilde{\boldsymbol{A}} \widehat{\boldsymbol{U}}_{\boldsymbol{Y}} = \widehat{\boldsymbol{U}}_{\boldsymbol{Y}}^{\top} \boldsymbol{Y} \boldsymbol{V}_{\boldsymbol{\Omega}} \Sigma_{\boldsymbol{\Omega}}^{+} \boldsymbol{U}_{\boldsymbol{\Omega},1}^{\top} \widehat{\boldsymbol{U}}_{\boldsymbol{Y}}, \qquad \widetilde{\boldsymbol{B}}_{\mathrm{R}} = \widetilde{\boldsymbol{B}} = \widehat{\boldsymbol{U}}_{\boldsymbol{Y}}^{\top} \boldsymbol{Y} \boldsymbol{V}_{\boldsymbol{\Omega}} \Sigma_{\boldsymbol{\Omega}}^{+} \boldsymbol{U}_{\boldsymbol{\Omega},2}^{\top}, \tag{46}$$

which maps the current reduced state $\boldsymbol{x}_{\mathrm{R}}(t_i)$ and actuation $\boldsymbol{u}(t_i)$ to the next reduced state $\boldsymbol{x}_{\mathrm{R}}(t_{i+1})$. It can be verified easily that if we use the truncated SVD (as defined by 5), instead of the full SVD, for $\boldsymbol{\Omega}$ in (45) and follow the similar steps afterward, we get an approximation of (46):

$$\widehat{\boldsymbol{A}}_{\mathrm{R}} = \widehat{\boldsymbol{U}}_{\boldsymbol{Y}}^{\top} \boldsymbol{Y} \widehat{\boldsymbol{V}}_{\boldsymbol{\Omega}} \widehat{\Sigma}_{\boldsymbol{\Omega}}^{-1} \widehat{\boldsymbol{U}}_{\boldsymbol{\Omega},1}^{\top} \widehat{\boldsymbol{U}}_{\boldsymbol{Y}} = \boldsymbol{A}_{\mathrm{R,DMDc}}; \qquad \widehat{\boldsymbol{B}}_{\mathrm{R}} = \widehat{\boldsymbol{U}}_{\boldsymbol{Y}}^{\top} \boldsymbol{Y} \widehat{\boldsymbol{V}}_{\boldsymbol{\Omega}} \widehat{\Sigma}_{\boldsymbol{\Omega}}^{-1} \widehat{\boldsymbol{U}}_{\boldsymbol{\Omega},2}^{\top} = \boldsymbol{B}_{\mathrm{R,DMDc}}.$$

In summary, the aforementioned method can be carried out using gradient descent-based optimization and leads to the same ROM as DMDc, when $\boldsymbol{E}_{\boldsymbol{x}} = \widehat{\boldsymbol{U}}_{\boldsymbol{Y}}^{\top}$. However, in this method, the benefit of dimensionality reduction is realized only when linear networks are used. A nonlinear counterpart (a DNN in the context of this paper) of $\widetilde{\boldsymbol{A}}_{\mathrm{R}}$, i.e., a nonlinear mapping from $\mathbb{R}^{r_{\boldsymbol{x}}}$ to $\mathbb{R}^{r_{\boldsymbol{x}}}$, cannot be pre-computed from a nonlinear counterpart of $\widetilde{G}$, unlike the linear case (46). Consequently, we lose the benefit of dimensionality reduction when nonlinear networks are used.

## C DETAILS ON REACTION−DIFFUSION SYSTEM EXPERIMENT

### C.1 SYSTEM DEFINITION

The Newell–Whitehead–Segel reaction-diffusion equation with the Neumann boundary condition is defined by

$$\frac{\partial q}{\partial t} = \sigma \nabla^2 q + q(1 - q^2) + \mathbf{1}_{\mathbb{W}} w \quad \text{in } \mathbb{I} \times \mathbb{R}^{+},$$
$$\nabla q(\zeta_l, t) = \nabla q(\zeta_r, t) = 0, \quad t \in \mathbb{R}^{+}. \tag{47}$$

In (47), $q(\zeta, t) \in \mathbb{R}$ denotes the measurement variable such as concentration or temperature at location $\zeta \in \mathbb{I} \subset \mathbb{R}$ and time $t$; $\sigma$ denotes the diffusion coefficient; $w(t) \in \mathbb{R}$ is the actuation at time $t$ and $\mathbf{1}_{\mathbb{W}}(\zeta)$ is the indicator function with $\mathbb{W} \subset \mathbb{I}$; $\zeta_l$ and $\zeta_r$ denote the boundary points of $\mathbb{I}$. We use $\mathbb{I} = (-1, 1), \mathbb{W} = (-0.2, 0.2)$, and $\sigma = 0.2$.

### C.2 DATASET

We use FEniCS (Logg et al. (2012)), an open-source computing platform for solving PDEs using the finite element method, with Python interface to generate the dataset. For the reaction-diffusion system of (47), we generate 100 training sequences of length 50 with time step size 0.01 and 256 nodes in $\mathbb{I}$. The initial conditions and actuations of these sequences are given by

$$q(\zeta, 0) = |a| \sum_{k=0}^{4} b_k T_k(\zeta), \quad \zeta \in \mathbb{I}, \tag{48}$$

and

$$w(t_i) = 10 g_i \max_{\zeta} |q(\zeta, t_{i-1})|, \quad i = 1, 2, \cdots, 49, \tag{49}$$

where $T_k$ denotes the $k^{\mathrm{th}}$ Chebyshev polynomial of the first kind, and $a \sim \mathcal{N}(0, 1)$, $b_k, g_i \sim \mathcal{U}(-1, 1)$ are chosen randomly. Similarly, 100 sequences are generated for the test set to evaluate the prediction performance.

### C.3 DNN ARCHITECTURES

Figure 4 shows the DNN architectures used for different modules in the reaction–diffusion experiment. The state encoder comprises 1D convolutional layers, followed by fully connected layers. The state decoder has the reversed order with convolutional layers replaced by transposed

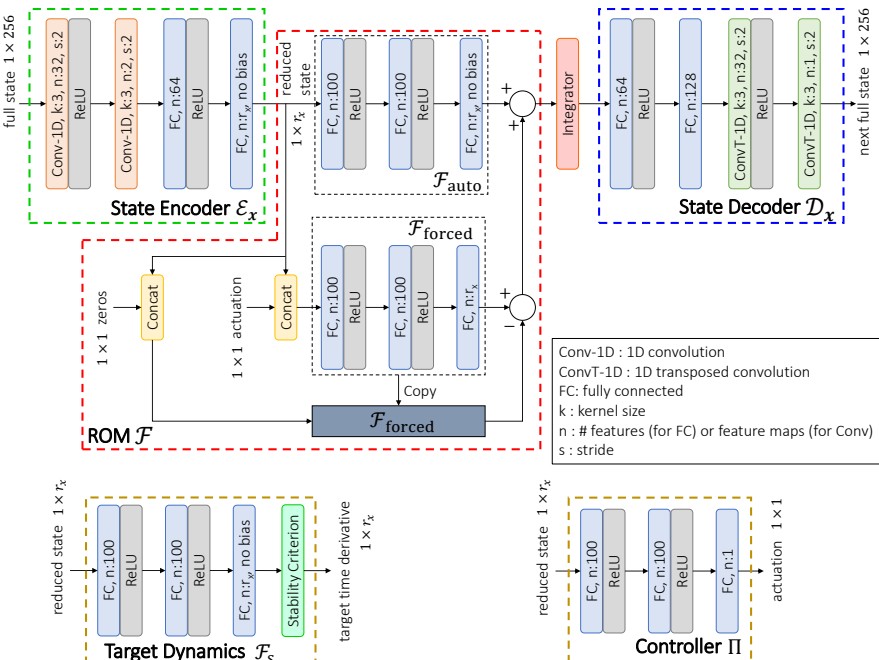

Figure 4: Architectures for all the DNN modules used in the reaction–diffusion experiment. The 'Copy' operation denotes the reuse of the same DNN block for zero and nonzero actuation. The 'Concat' operator concatenates the input features along the last dimension. Zeros are concatenated to the reduced state to evaluate the component $\mathcal{F}_{\text{forced}}(\boldsymbol{x}_{\text{R}}, \boldsymbol{0})$. The 'Integrator' performs the numerical integration to get the next state as mentioned in 4.1.3. The 'Stability Criterion' block implements (17).

convolutional layers. The ROM is designed by breaking the function $\mathcal{F}$ into two components: $\mathcal{F}(\boldsymbol{x}_{\text{R}}, \boldsymbol{u}_{\text{R}}) = \mathcal{F}_{\text{auto}}(\boldsymbol{x}_{\text{R}}) + \mathcal{F}_{\text{forced}}(\boldsymbol{x}_{\text{R}}, \boldsymbol{u}_{\text{R}}) - \mathcal{F}_{\text{forced}}(\boldsymbol{x}_{\text{R}}, \boldsymbol{0})$. $\mathcal{F}_{\text{auto}}$ represents the autonomous dynamics that does not depend on the actuation, whereas $\mathcal{F}_{\text{forced}}$ is responsible for the impact of actuation on dynamics. The composition $\mathcal{F}_{\text{forced}}(\boldsymbol{x}_{\text{R}}, \boldsymbol{u}_{\text{R}}) - \mathcal{F}_{\text{forced}}(\boldsymbol{x}_{\text{R}}, \boldsymbol{0})$ ensures that the component responsible for learning the impact of actuation on the dynamics provides nonzero output only when the actuation is nonzero. Two multilayer perceptions (MLPs) are used to implement $\mathcal{F}_{\text{auto}}$ and $\mathcal{F}_{\text{forced}}$. This specific structure of the ROM is not crucial and a single neural network representing $\mathcal{F}(\boldsymbol{x}_{\text{R}}, \boldsymbol{u}_{\text{R}})$ works as well. However, we observe better performance in experiments when the aforementioned structure is used. The output of the ROM is integrated using a numerical integrator to get the next state. The controller is implemented using an MLP. The target dynamics is implemented using another MLP, followed by a stability criterion in the form of (17).

## C.4 TRAINING SETTINGS

We use $r_{\boldsymbol{x}} = 5$ in the prediction task and $r_{\boldsymbol{x}} = 2$ in the control task for all the methods. All modules are implemented in PyTorch. In both of the learning phases, learning ROM and learning controller, we use the Adam optimizer with an initial learning rate of $0.001$ and apply an exponential scheduler with a decay of $0.99$. Modules are trained for $100$ epochs in mini-batches of size $32$. $10\%$ of the training data is used for validation to choose the best set of models. For DeepROM training, we use $\beta_2 = 1$ in (14). For learning control, we use $\beta_3 = 0.2$ in (16), $\alpha = 0.2$ in (17), and $\boldsymbol{K} = 0.5\boldsymbol{I}_{r_{\boldsymbol{x}}}$ in (18). Since the learned ROMs from one training instance to another can vary, the hyperparameter pair $(\alpha, \beta_3)$ may require re-tuning accordingly.

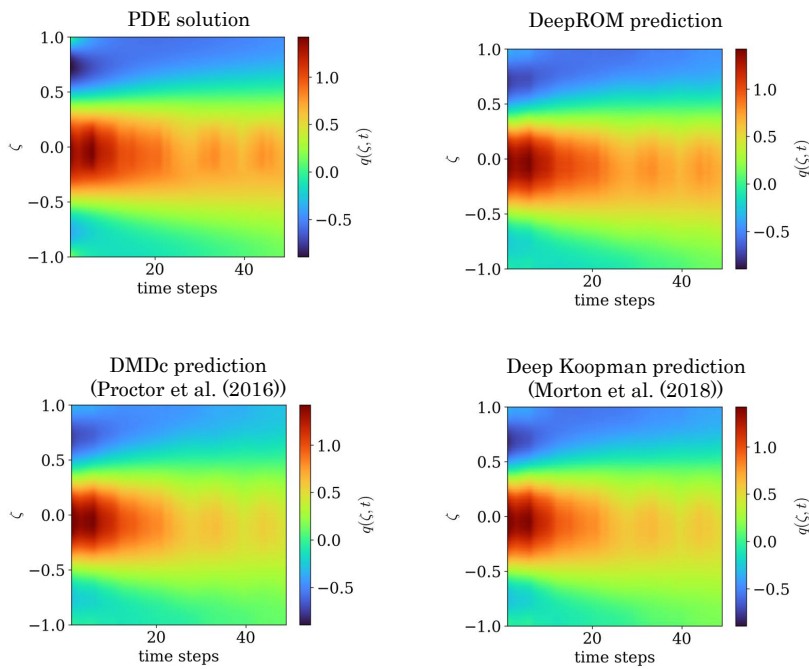

Figure 5: Qualitative comparison of prediction performance for DMDc, Deep Koopman, and Deep-ROM in the reaction–diffusion example using one example sequence.

### C.5 ADDITIONAL RESULTS

Figure 5 shows the visual comparison of the recursive multi-step predictions obtained using DMDc, Deep Koopman model, and DeepROM. The color maps are shown for one example sequence with one training instance.

Figure 6 visually compares the uncontrolled solution and the controlled solutions obtained using the three methods. When uncontrolled, the system reaches the stable equilibrium at 1, whereas the feedback-controlled system is stabilized at the desired state 0 in all cases.

Figure 7 compares the prediction and control performance of PCOL (Hwang et al. (2022)) and Deep-ROC. The surrogate model for PCOL shares a similar architecture as DeepROM except the latent dynamic model uses a discrete-time formulation. Also, the model is trained with the loss functions as proposed by Hwang et al. (2022). PCOL optimizes the control input of the trained model through backpropagation. Such optimization technique is computationally expensive for time-dependent PDEs, particularly when a long trajectory is needed to be rolled out. We observed that we can optimize the control input only up to a certain time step. Though the system state reaches the target within this timeframe, it fails to stay stable since the target is an unstable equilibrium and the control input is no longer effective.

### D DETAILS ON VORTEX SHEDDING SUPPRESSION EXPERIMENT

#### D.1 SYSTEM DEFINITION

The dynamics is governed by the incompressible Navier-Stokes equations given by

$$\frac{\partial \boldsymbol{v}}{\partial t} - \nu \nabla^2 \boldsymbol{v} + (\boldsymbol{v} \cdot \nabla)\boldsymbol{v} = -\frac{1}{\rho}\nabla p + \mathbf{1}_{\mathbb{W}}\boldsymbol{w}, \quad \nabla \cdot \boldsymbol{v} = \mathbf{0} \quad \text{in } \mathbb{I} \times \mathbb{R}^+, \tag{50}$$

where $\boldsymbol{v}(\boldsymbol{\zeta}, t) \in \mathbb{R}^2$ denotes the flow velocity at location $\boldsymbol{\zeta} \in \mathbb{I} \subset \mathbb{R}^2$ and time $t$, $p(\boldsymbol{\zeta}, t) \in \mathbb{R}$ denotes the pressure, $\nu$ denotes the kinematic viscosity and $\rho$ denotes the density of the fluid. $\boldsymbol{w}(\boldsymbol{\zeta}, t)$ is the actuation/force applied to the system and $\mathbf{1}_{\mathbb{W}}(\boldsymbol{\zeta})$ is the indicator function with $\mathbb{W} \subset \mathbb{I}$. We use

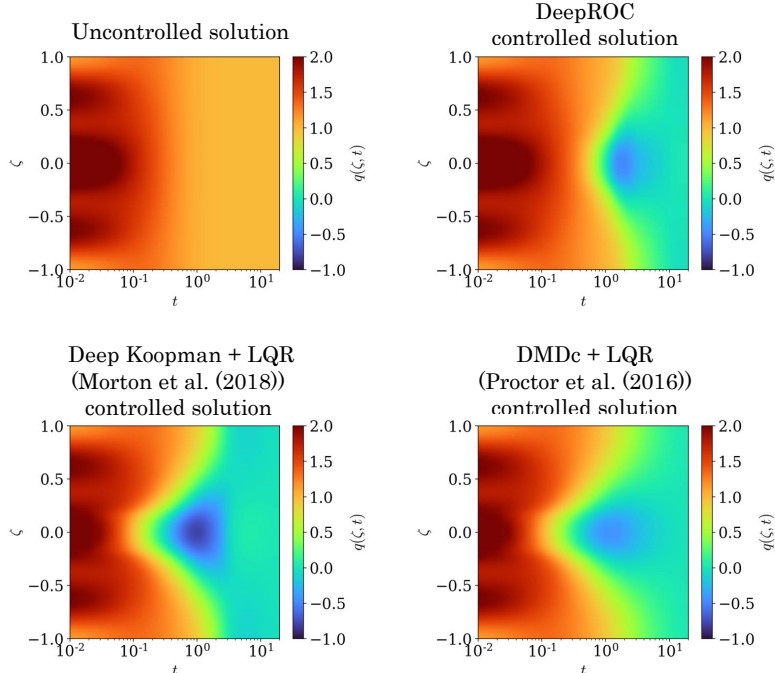

Figure 6: Visual comparison of the uncontrolled solution and the controlled solutions of the reaction–diffusion system using DeepROC, Deep Koopman + LQR, and DMDc + LQR.

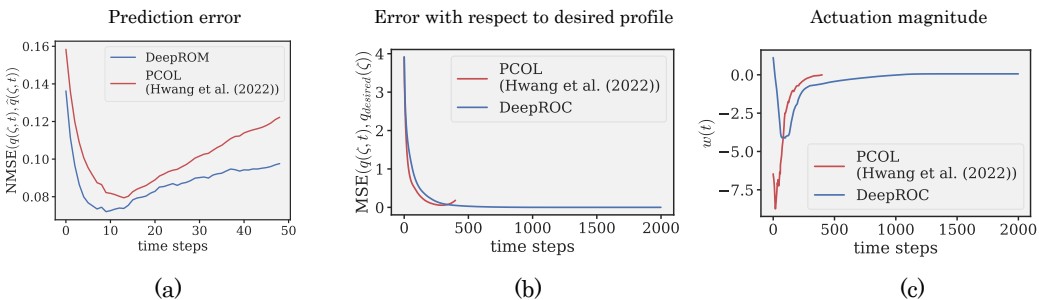

Figure 7: (a): Prediction performance of PCOL (Hwang et al. (2022)) and DeepROM in the reaction–diffusion example. (b,c): Control performance of PCOL and DeepROC in the reaction–diffusion example.

$\mathbb{I} = (0, 2.2) \times (0, 0.41)$ and $\mathbb{W} = (0.11, 0.77) \times (0, 0.41)$. We use the domain $\mathbb{W}$ for observation and distributed actuation. The Stokes flow is used as the desired state for the control task.

## D.2 DATASET

For the flow past a circular cylinder problem, the geometry and physical parameters of the system are taken from the DFG 2D-2 benchmark (Schäfer et al. (1996)). The geometry is shown in Figure 8. We use the blue-shaded region for observation and actuation. Following the DFG 2D-2 benchmark, we use the no-slip boundary condition of zero velocity for the walls and the cylinder boundary, zero outlet pressure, and the inflow velocity profile (at the inlet) as

$$\boldsymbol{v}(\boldsymbol{\zeta}, t) = \left(1.5 \frac{4\zeta_2(0.41 - \zeta_2)}{0.41^2}, 0\right),$$
(51)

where $\zeta_1$ and $\zeta_2$ denote the horizontal and vertical coordinates, respectively, of $\zeta$. We use kinematic viscosity $\nu = 0.002$ and density $\rho = 1$ leading to the Reynolds number $Re = 50$. The training sequence of length 5000 is generated in FEniCS with a time step size 0.001 and applying actuations

$$\boldsymbol{w}(\boldsymbol{\zeta}, t) = a \sum_{k=0}^{4} \left[ \sin(k\pi(\zeta_1 - 0.11)/0.66) \quad \sin(k\pi\zeta_2/0.41) \right] \begin{bmatrix} b_{k,1,1} & b_{k,2,1} \\ b_{k,1,2} & b_{k,2,2} \end{bmatrix}, \quad \boldsymbol{\zeta} \in \mathbb{W}, \quad (52)$$

where $a \sim \mathcal{U}(0, 1)$ and $b_{k,i,j} \sim \mathcal{U}(-1, 1), i, j = 1, 2$ are chosen randomly. Similarly, a test sequence is generated to evaluate the prediction performance. For learning control, we use the Stokes flow or creeping flow as the desired state, which can be obtained by solving the Stokes equations

$$\nu \nabla^2 \boldsymbol{v} - \frac{1}{\rho} \nabla p = \boldsymbol{0}, \quad \nabla \cdot \boldsymbol{v} = \boldsymbol{0} \quad \text{in } \mathbb{I} \times \mathbb{R}^+. \quad (53)$$

For training, the flow velocity data from the observation region (blue shaded in Figure 8) are interpolated onto a rectangular uniform grid of size $32 \times 48$ so that it can be used in standard CNNs.

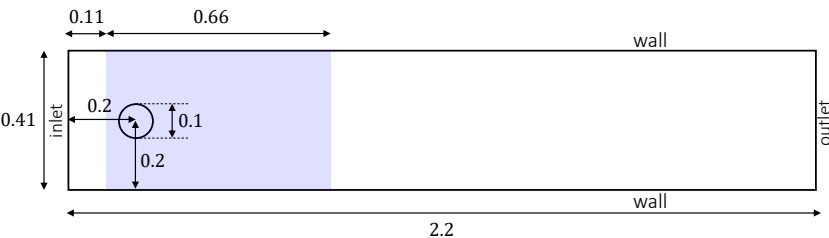

Figure 8: Geometry of the flow past a circular cylinder set-up.

### D.3 DNN ARCHITECTURES

Figure 9 shows the DNN architectures used for different modules in the vortex shedding control experiment. The architectures for the ROM and target dynamics are the same as in the previous example. Moreover, the state encoder and decoder have similar architectures as the previous example except for the 1D convolutions and transposed convolutions are replaced by their 2D counterparts. Here, an additional module is used: the control encoder for encoding the distributed control/actuation. It has the same architecture as the state encoder. To learn the distributed actuation, we design the controller as a linear combination of space-dependent polynomial basis functions. One MLP is used to learn these space-dependent polynomial basis functions given the locations of the actuation nodes and another MLP is used to learn the corresponding coefficients. The actuation is computed as the dot product of the polynomial basis terms and the coefficient vector. We use this architecture instead of a standard convolutional one because the PDE solver takes the actuation input in a triangular mesh, not in a uniform rectangular grid. The polynomial basis architecture can be used to compute actuation in both uniform rectangular grid during training and triangular mesh during evaluation.

### D.4 TRAINING SETTINGS

We use $r_{\boldsymbol{x}} = 5$ in both the prediction task and control task for all the methods. All modules are implemented in PyTorch. In both of the learning phases, learning ROM and learning controller, we use the Adam optimizer with an initial learning rate of 0.001 and apply an exponential scheduler with a decay of 0.99. Modules are trained for 100 epochs in mini-batches of size 32. 10% of the training data is used for validation to choose the best set of models. For DeepROM training, we use $\beta_2 = 1$ in (14). For learning control, we use $\beta_3 = 2$ in (16), $\alpha = 0.1$ in (17), and $\boldsymbol{K} = 0.5\boldsymbol{I}_{r_{\boldsymbol{x}}}$ in (18). Since the learned ROMs from one training instance to another can vary, the hyperparameter pair $(\alpha, \beta_3)$ may require re-tuning accordingly.

### D.5 ADDITIONAL RESULTS

Figure 10 shows the visual comparison of the recursive multi-step predictions obtained using DMDc, Deep Koopman model, and DeepROM. Unlike DeepROM, DMDc and Deep Koopman model are

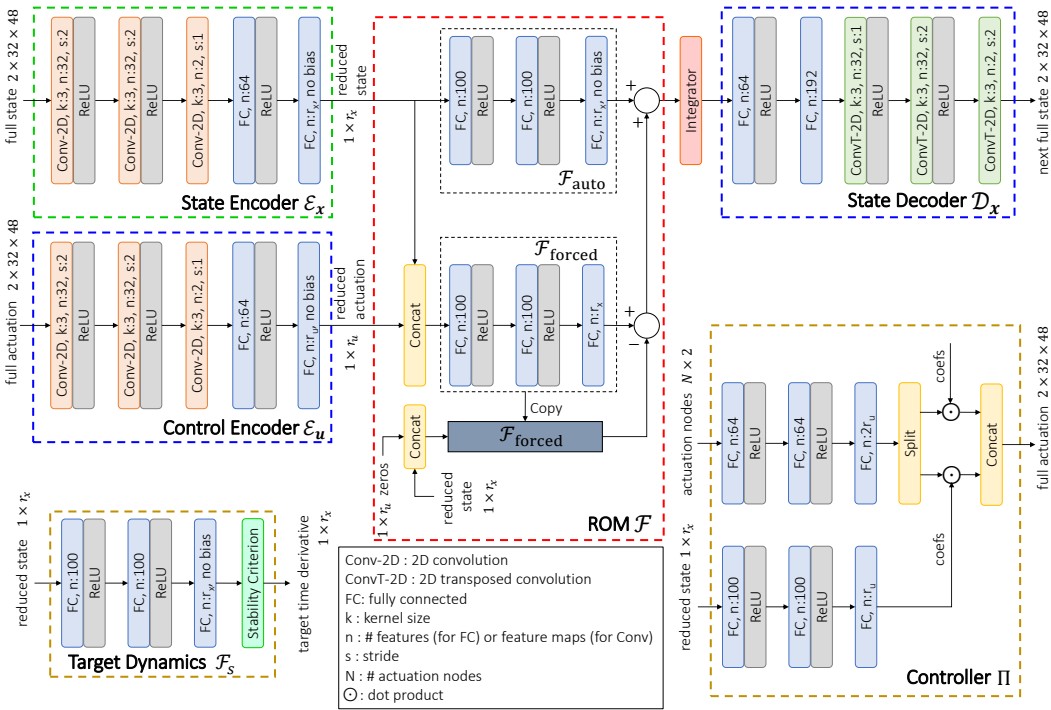

Figure 9: Architectures for all the DNN modules used in the fluid flow experiment. The 'Split' operator splits the input features into two vectors, along the last dimension. These split vectors represent the space-dependent polynomial basis associated with the horizontal and vertical components of the actuation.

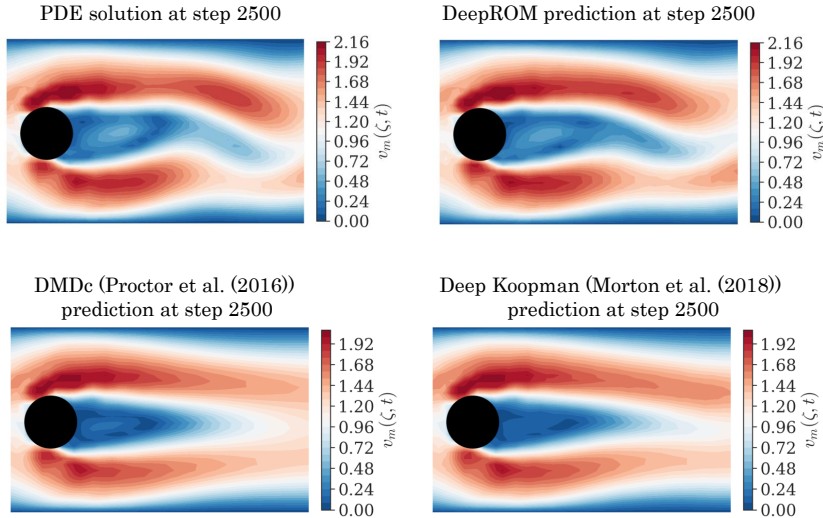

Figure 10: Qualitative comparison of prediction performance for DMDc, Deep Koopman, and Deep-ROM in the fluid flow example. Predictions at time step 2500 for the test sequence are visually compared with the solution from a PDE solver. $v_m$ denotes the velocity magnitude.

unable to capture the shedding pattern in multi-step prediction as shown in the contour plots of the velocity magnitude.

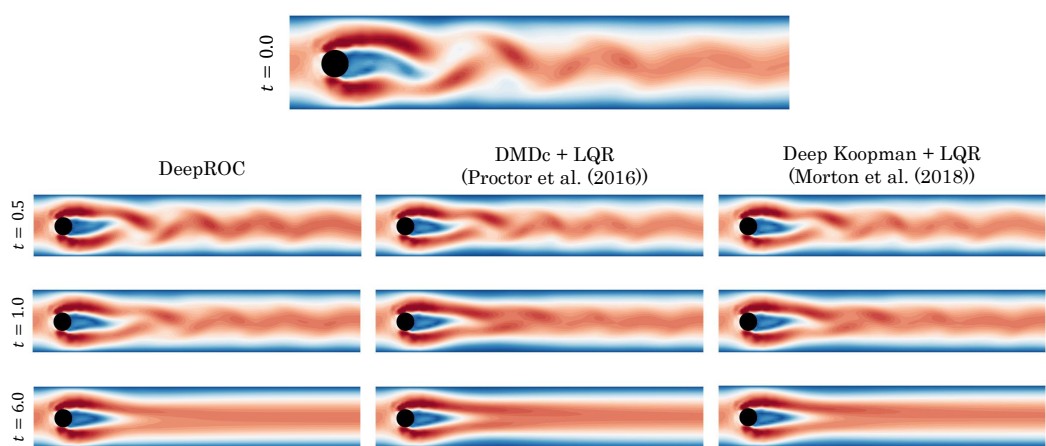

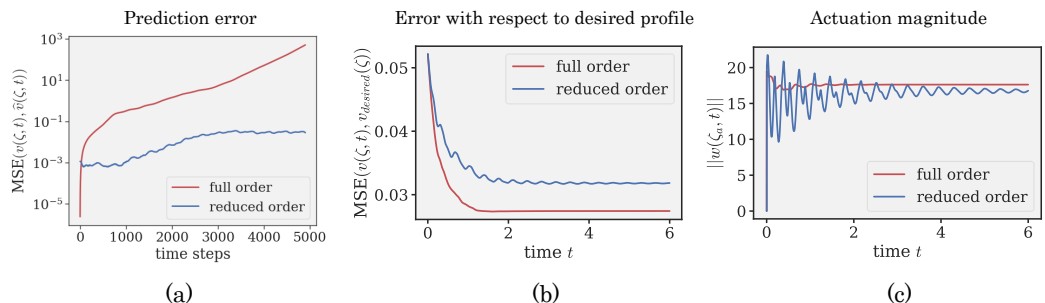

Figure 11: Visual comparison of the velocity magnitude of the flow over time subjected to the controllers obtained using DeepROC, Deep Koopman + LQR, and DMDc + LQR.

Figure 12: (a): Prediction performance of FOM and DeepROM in the fluid flow example. (b, c): Control performance of FOM + NI4C (Saha et al. (2021)) and DeepROC in the vortex shedding suppression task.

Figure 11 shows the velocity magnitude of the controlled flow for DeepROC, Deep Koopman + LQR, and DMDc+LQR at different times, starting from a von Kármán vortex street pattern. All methods accomplish a similar steady-state flow pattern where vortex shedding has been suppressed.

Figure 12 shows the performance comparison with full-order model-based prediction and control. The full-order model (FOM) removes the bottleneck FC layers from the encoder and decoder and uses CNNs for the dynamic model instead of MLPs.

Table 1: Computational costs of FOM and DeepROM

| Model | | #Params | #FLOPs | NFE* | #E-FLOPs** |
|-------|--------|---------|--------|-------|-----------|
| FOM | | 76.93K | 22.34M | 42.83 | 956.80M |
| DeepROM | Encoder | 23.11K | 1.17M | N/A | |
| | ROM | 22.91K | 34.00K | 13.13 | 6.11M |
| | Decoder | 23.11K | 4.49M | N/A | |

\* NFE: Avg. number of function evaluations by the numerical integrator
\*\* #E-FLOPs(FOM) = #FLOPs(FOM) × NFE(FOM)
   #E-FLOPs(DeepROM) = #FLOPs(Enc) + #FLOPs(Dec) + #FLOPs(ROM) × NFE(ROM)

Figure 13 shows the prediction performance comparison with VideoGPT (VQ-VAE + Transformer) for an unforced system. VideoGPT generates accurate predictions for the time window it is trained with. However, as seen from Figure 13, prediction accuracy drops sharply after each recursive prediction window.

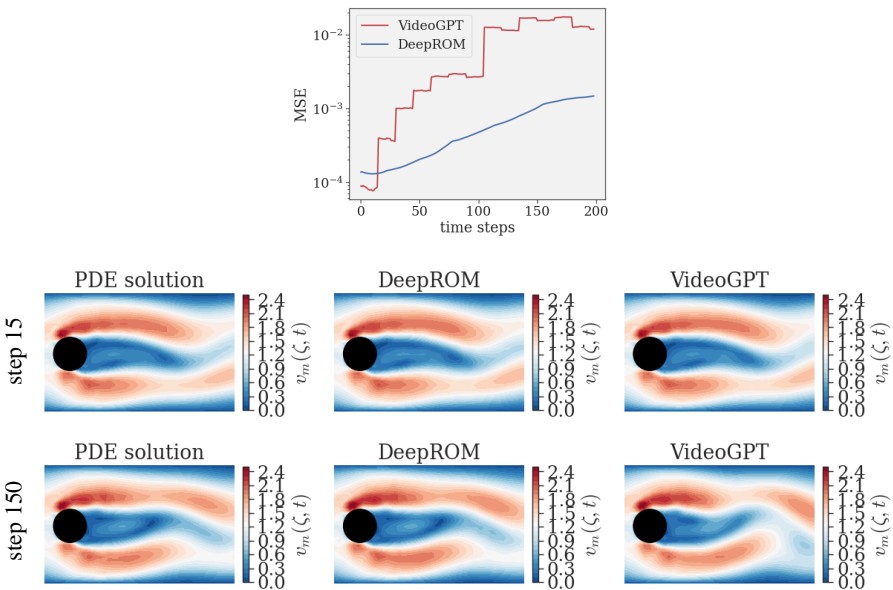

Figure 13: Prediction performance of VideoGPT and DeepROM for unforced fluid flow.

## E  EMPIRICAL SIMILARITY BETWEEN DMDc AND LAROM

$L_{\text{recon}}$, as defined in 4.1.2, is minimized for any invertible matrix $C$, $D_x = \widehat{U}_Y C$, and $E_x = C^{-1}\widehat{U}_Y^\top$. When optimized using gradient descent, it is highly unlikely to get $C$ as the identity matrix like DMDc. Rather, we expect a random $C$. Therefore, we need additional constraints to demonstrate empirical similarity with DMDc. For this purpose, we tie the matrices $E_x$ and $D_x$ to be the transpose of each other and add a semi-orthogonality constraint $\beta_4 \|E_x E_x^\top - I_{r_x}\|, \beta_4 > 0$ to the optimization objective of (13).

The dynamic modes for LAROM are computed as $\varphi_i = D_x z_i$, where $z_i$ is the $i^{\text{th}}$ eigenvector of $A_{\text{R}}$. Similarly, the dynamic modes for DMDc are computed as $\varphi_{i,\text{DMDc}} = D_{\text{DMDc}} z_{i,\text{DMDc}}$, where $z_{i,\text{DMDc}}$ is the $i^{\text{th}}$ eigenvector of $A_{\text{R,DMDc}}$. Note, these dynamic modes are similar to the ones used in the original DMD algorithm Schmid (2010), not the exact modes obtained in Proctor et al. (2016). Exact modes cannot be computed for LAROM since it does not involve SVD. Modes defined by $\varphi_{i,\text{DMDc}} = D_{\text{DMDc}} z_{i,\text{DMDc}} = \widehat{U}_Y z_{i,\text{DMDc}}$ are the orthogonal projection of the exact modes onto the range of $Y$ (Theorem 3, Tu et al. (2014)).

Figure 14 compares the dynamic modes of the reaction-diffusion system, obtained using DMDc and LAROM for the case when the dimension of the ROMs is 3. It is important to note that the numbering of the modes is arbitrary as the optimal ranking of DMDc modes is not trivial. The correspondence between the DMDc modes and LAROM modes are determined by comparing the eigenvalues of $A_{\text{R,DMDc}}$ and $A_{\text{R}}$. Dynamic modes of both methods are similar except for the different signs of the first two modes.

Figure 15 compares the first two oscillatory dynamic modes obtained using DMDc and LAROM for the fluid system. Only the streamwise components are shown for brevity. Also, complex modes occur in conjugate pairs and only one from each pair is shown. The correspondence between the DMDc modes and LAROM modes are determined by comparing the eigenvalues of $A_{\text{R,DMDc}}$ and $A_{\text{R}}$. Dynamic modes identified by LAROM are similar to the ones obtained from DMDc, except the real and imaginary components of the first mode are swapped.

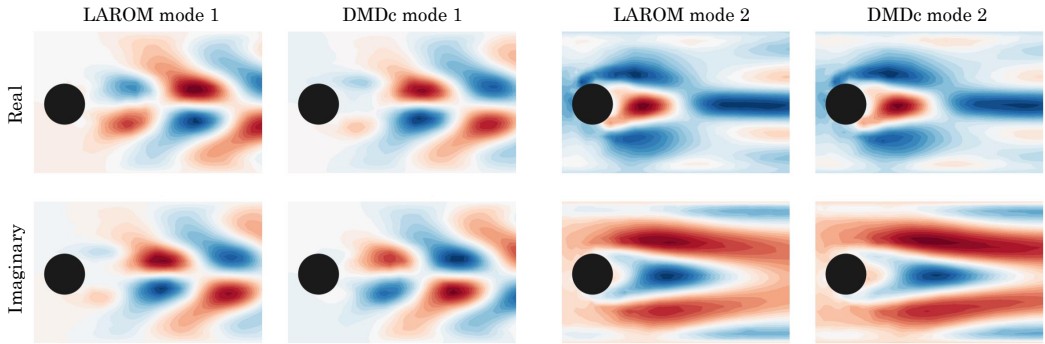

Figure 14: The first three dynamic modes of the reaction–diffusion system, obtained using DMDc and LAROM.

Figure 15: The first two dynamic modes obtained using DMDc and LAROM for the flow past a cylinder system.

## F   ARCHITECTURE AND TRAINING DETAILS FOR THE DEEP KOOPMAN MODEL

For the encoder and decoder of the Deep Koopman model, we use the same architectures as our state encoder and state decoder. As mentioned in section 5.1, we consider both the system and input matrices of the ROM to be fixed during operation, in contrast to the original method proposed by Morton et al. (2018). Therefore, during training, these matrices are treated as trainable global parameters. Similar to Morton et al. (2018), the input matrix is optimized by gradient descent during training along with the encoder-decoder parameters, whereas the system matrix is obtained using linear least-squares regression. The datasets are divided into staggered 32-step sequences for training, and the model is trained by generating recursive predictions over 32 steps following Morton et al. (2018). We train the model using the Adam optimizer with an initial learning rate of $0.001$ and an exponential decay of $0.99$ for $200$ epochs in mini-batches of size $8$. $10\%$ of the training data is used for validation to choose the best set of models.

As mentioned in 5.3, we utilize a low-dimensional representation of the distributed actuation for Deep Koopman + LQR, instead of directly estimating the high-dimensional actuation. The distributed actuation is represented as a linear combination of the same space-dependent sinusoidal basis functions used for dataset generation, which are given by (52). The controller is designed to estimate the coefficients $b_{k,i,j}; i, j = 1, 2; 0 \leq k \leq 4$.

