# OpenReview forum: "Dynamic Mode Decomposition-inspired Autoencoders for Reduced-order Modeling and Control of PDEs : Theory and Design"
_ICLR.cc/2024/Conference — Submitted to ICLR 2024_

### Official Review · Reviewer_Wa1W · 2023-10-25

**Soundness:** 4 excellent
**Presentation:** 3 good
**Contribution:** 3 good
**Rating:** 6
**Confidence:** 3

**Summary:**

The paper presents a deep autoencoding learning method for modeling and controlling spatiotemporal systems driven by partial differential equations (PDEs). The authors introduce a linear autoencoding model, then expand to a nonlinear framework using deep autoencoding. This framework aids in designing controllers using stability-constrained neural networks, without needing prior system equation knowledge. Its efficacy is demonstrated on reaction-diffusion and fluid flow systems using time series data. Empirical results are promising, when compared to existing DMD based methods.

**Strengths:**

This is a theoretically sound paper for learning a reduced order model for PDE systems. Empirical results are very promising when compared to existing DMD based methods.

**Weaknesses:**

The paper only compared to DMD based methods, however, transformer based VAEs like VQ-VAEs and VQ-GANs have achieved SOTA performances on video data. It would be very beneficial to see a comparison with these more data-driven methods.

**Questions:**

N/A

---

> ### Author Response · Authors · 2023-11-21
> **Response to reviewer Wa1W**
>
> **Q1. The paper only compared to DMD-based methods, however, transformer-based VAEs like VQ-VAEs and VQ-GANs have achieved SOTA performances on video data. It would be very beneficial to see a comparison with these more data-driven methods.**
>
> **A1.** We appreciate your suggestion to explore transformer-based VAEs for comparison in our study. However, application of these methods to controlled dynamical systems with dynamic control inputs is not straightforward and requires significant modification to the architecture. Recognizing this, we consider such models for our fluid flow example without any control inputs and investigate the prediction performance of VideoGPT (VQ-VAE + Transformer). Our experiment shows that VideoGPT generates accurate predictions in the short term but falls short in capturing long-term dynamics. We value your insightful suggestion and have included quantitative and qualitative results in the supplementary to encourage potential future research directions.

---

> > ### Author Response · Authors · 2023-11-22
> > **Gentle reminder**
> >
> > Thanks again for your constructive feedback and comments. We have updated the paper accordingly. As the discussion period ends soon, we would appreciate knowing if our response addressed your concerns.

---

> > ### Comment · Reviewer_Wa1W · 2023-11-23
> >
> > Thanks for the response. The new results look good, and I will raise my score to 6 accordingly.

---

> > > ### Author Response · Authors · 2023-11-23
> > > **Thanks to reviewer Wa1W**
> > >
> > > Thank you for acknowledging the recent updates and for raising the score. Your recognition of our efforts serves as a significant encouragement to us. We deeply appreciate your constructive feedback, which has immensely contributed to improving the quality of our work. Thanks once again for your invaluable input.

---

### Official Review · Reviewer_9vtz · 2023-10-26

**Soundness:** 3 good
**Presentation:** 4 excellent
**Contribution:** 3 good
**Rating:** 6
**Confidence:** 3

**Summary:**

The authors propose a framework for autoencoder-based modeling and control learning for PDE-driven dynamical systems. Traditional linear encoder and decoder are replaced by nonlinear parametrization with neural networks. The dynamic is also modeled by a neural network. The authors present numerical examples to validate their method.

**Strengths:**

The presentation of the paper is clear. I am not an expert in this area but I can clearly understand this work.
In the numerical results, the authors compare their method with existing methods. The method proposed in this paper has similar error performance but less cost due to a dimension reduction.

**Weaknesses:**

The idea of replacing linear encoder and decoder with neural networks is simple. There is not much innovation in this paper.

**Questions:**

About the learning of the control. Since the control is of feedback type, can we use a simple supervised learning from the observed data instead of loss (15)?
Page 6 middle. In the integration, it should be t instead of t_i?

---

> ### Author Response · Authors · 2023-11-21
> **Response to reviewer 9vtz**
>
> **Q1. The idea of replacing linear encoder and decoder with neural networks is simple. There is not much innovation in this paper.**
>
> **A1.** While we appreciate your assessment, it is important to note that the replacement of linear encoders and decoders with neural networks is not the primary focal point of our contribution. We acknowledge the existence of prior works exploring similar model configurations within the field.
>
> Our paper aims to establish a theoretical connection between the traditional linear approach and the emerging deep learning-based methods for the control of dynamical systems. There are several viable approaches to constructing and training an autoencoder-based ROM for PDE-driven dynamical systems, as demonstrated in recent literature. In our work, the architecture and loss functions are strategically formulated to forge a bridge between the conventional linear approach and DNN-based methods, specifically through the lens of a linear autoencoding representation of DMDc.
>
> **Q2. About the learning of the control. Since the control is of feedback type, can we use a simple supervised learning from the observed data instead of loss (15)?**
>
> **A2.** The data used for training the model encompass observations obtained under random control inputs. We assume that we do not have any trajectory data under an effective control policy. Without such data formulating supervised learning methods becomes challenging. We would appreciate additional elaboration on this comment, as it would help us understand the feasibility of employing supervised learning.
>
> **Q3. In the integration, it should be t instead of $t_i$?**
>
> **A3.** Yes, it should be $t$. Thank you for pointing out the error.

---

> > ### Author Response · Authors · 2023-11-22
> > **Gentle reminder**
> >
> > Thanks again for your constructive feedback and comments. We have updated the paper accordingly. As the discussion period ends soon, we would appreciate knowing if our response addressed your concerns.

---

### Official Review · Reviewer_WjTs · 2023-11-02

**Soundness:** 2 fair
**Presentation:** 3 good
**Contribution:** 3 good
**Rating:** 6
**Confidence:** 3

**Summary:**

This paper devises a new method to develop ROMs for both linear and nonlinear control problems using autoencoding techniques. To guarantee its performance, the model encourages DMDc-like solutions on linear problems by designing a loss taking both dynamics prediction error and reconstruction error into consideration whereas nonlinear ones are tackled similarly by replacing all trainable components in the linear version into DNN.

**Strengths:**

1) Clarity: The exposition is basically clear. The description of the proposed method, the datasets involved and the baselines are detailed.
2) Quality: The math derivation seems sound.

**Weaknesses:**

1) Lack of ablation study. The paper mainly focuses on a new reduced-order approach (DeepROM) to model the system’s dynamics and adopts an existing method NI4C(Saha et al., 2021) to learn the control. Thus in the experiment, the original full-order NI4C should be compared, as an ablation study for DeepROM. However, in current experiments, only LQR controls are compared.

2) Insignificant performance improvement. The performance of the DeepROC on the vortex-shedding task seems to be worse than LQR baselines (Figure 3(b,c)).  The overall idea of reduced-order control is not new, thus the performance improvement is an important measure of contribution.

3) Lack of higher dimensional and more complex datasets. In the abstract, the authors state that: "Controlling complex PDEs often necessitates dimensionality reduction for computational efficiency." However, the experiments are about 1D reaction-diffusion and 2D N-S(with low Re). Such systems can be learned and predicted efficiently via full-order neural networks, such as simple FNN, DeepONet, FNO, PINN, and their variants. More complex datasets, such as 2D/3D N-S with turbulence or shock are needed to show the necessity of reduced-order models.

**Questions:**

The major questions are listed in the Weakness part. Here are some minor questions about network details:

1) How to choose tunable hyperparameter $\beta_{1,2,3}$?
2) Does adding a decoder and a reconstruction loss for control $u$ affect performance?

---

> ### Author Response · Authors · 2023-11-21
> **Response to reviewer WjTs (1/2)**
>
> **Q1. Lack of ablation study. The paper mainly focuses on a new reduced-order approach (DeepROM) to model the system’s dynamics and adopts an existing method NI4C(Saha et al., 2021) to learn the control. Thus in the experiment, the original full-order NI4C should be compared, as an ablation study for DeepROM. However, in current experiments, only LQR controls are compared.**
>
> **A1.** Thank you for highlighting the importance of an ablation study, specifically comparing the original full-order NI4C method with our proposed reduced-order approach. We recognize the significance of this comparison in evaluating the efficacy of our method. We have added a comparison with full-order NI4C in the revised paper for the fluid flow experiment. While the full-order model (FOM) trained for NI4C exhibits superior prediction accuracy for the initial steps, it experiences a rapid decline in accuracy over time. In contrast, DeepROM provides accurate predictions for a much longer period of time. On the other hand, FOM + NI4C shows better performance for the control task. However, it is crucial to highlight that the advantage of employing a reduced-order model over a full-order model in control learning primarily resides in reduced computational complexity without a substantial compromise in accuracy. Effective FLOP count (#E-FLOPs) per training or prediction step for FOM is significantly higher (over 150X) than that of DeepROM despite a similar parameter count. The substantial computational disparity arises due to the NI4C’s requirement for a continuous-time formulation of the model, necessitating numerical integration during both training and inference.
>
> |Model| |#Params |#FLOPs| NFE| #E-FLOPs|
> |----------|----------|----------|----------|----------|----------|
> |FOM ||76.93K| 22.34M| 42.83| 956.80M|
> ||Encoder| 23.11K| 1.17M | N/A| |
> |DeepROM|ROM| 22.91K |34.00K |13.13 |6.11M
> ||Decoder| 23.11K| 4.49M |N/A|
>
> NFE: Avg. number of function evaluations by the numerical integrator
>
> #E-FLOPs(FOM) = #FLOPs(FOM) × NFE(FOM)
>
> #E-FLOPs(DeepROM) = #FLOPs(Enc) + #FLOPs(Dec) + #FLOPs(ROM) × NFE(ROM)
>
>
> **Q2. Insignificant performance improvement. The performance of the DeepROC on the vortex-shedding task seems to be worse than LQR baselines (Figure 3(b,c)). The overall idea of reduced-order control is not new, thus the performance improvement is an important measure of contribution.**
>
> **A2.** We acknowledge your concern regarding the observed performance of DeepROC on the vortex-shedding task compared to LQR baselines. We recognize that performance improvement would undoubtedly strengthen the contribution.  We explicitly acknowledge this limitation in the conclusion section.
>
> It is essential to clarify that our paper does not claim novelty in the general idea of reduced-order control. Instead, our focus was on establishing a theoretical connection between a traditional approach and the emerging deep learning-based methods for the control of dynamical systems.
>
>
> **Q3. Lack of higher dimensional and more complex datasets. In the abstract, the authors state that: "Controlling complex PDEs often necessitates dimensionality reduction for computational efficiency." However, the experiments are about 1D reaction-diffusion and 2D N-S(with low Re). Such systems can be learned and predicted efficiently via full-order neural networks, such as simple FNN, DeepONet, FNO, PINN, and their variants. More complex datasets, such as 2D/3D N-S with turbulence or shock are needed to show the necessity of reduced-order models.**
>
> **A3.** We agree that several existing DNN-based full-order models can efficiently predict the evolution of systems that we considered in this paper. However, many of these models are not straightforwardly applicable to systems with dynamic control inputs. Furthermore, learning control through a DNN-based full-order model is computationally expensive. Existing methods either optimize for static control inputs using backpropagation through the learned model or employ model predictive control to learn dynamic inputs. With full-order models, such optimizations with long trajectory rollouts are computationally expensive. For example, the FLOP count for $T$-step rollout for FNO is much higher than that of a discrete-time version of DeepROM for a large $T$.
>
> |Model| |#Params |#FLOPs| #FLOPs for $T$-step rollout|
> |----------|----------|----------|----------|----------|
> |FNO||926.85K| 8.37M| $(8.37 \times T)$M|
> ||Encoder| 23.11K| 1.17M ||
> |DeepROM|ROM| 11.20K| 11.00K| $(5.66 + 0.011 \times T)$ M|
> ||Decoder| 23.11K| 4.49M ||
>
> **Q4. How to choose tunable hyperparameter $\beta_{1,2,3}$?**
>
> **A4.** $\beta_{1,2}$ were chosen from a set of values by evaluating the prediction and reconstruction losses on a validation set. $\beta_3$ is chosen from a set of values by monitoring the Lyapunov energy along the trajectory of the ROM, under the learned policy.

---

> > ### Author Response · Authors · 2023-11-21
> > **Response to reviewer WjTs (2/2)**
> >
> > **Q5. Does adding a decoder and a reconstruction loss for control $u$ affect performance?**
> >
> > **A5.** The controller itself acts as a decoder. As mentioned in Appendix D.3 and shown in Fig. 9, one branch of the controller network learns space-dependent polynomial basis functions given the locations of the actuation nodes. The other branch maps the current reduced state to the coefficients for those basis functions. These coefficients can be considered as the latent code for control. However,  we did not observe any performance improvements by adding a reconstruction loss between these coefficients and the output of the control encoder.

---

> > > ### Comment · Reviewer_WjTs · 2023-11-22
> > >
> > > Thank you for the additional experiments and replies. I decided to raise my score to 6.
> > >
> > > The theoretical connection built in the paper seems solid, and the computational complexity analysis you provided in the rebuttal is clear. However, in my opinion, the performance drop and the lack of more complex experiments still limit the overall impact of the work.

---

> > > > ### Author Response · Authors · 2023-11-22
> > > > **Thanks to reviewer WjTs**
> > > >
> > > > Thank you for acknowledging the recent updates and for raising the score. Your recognition of our efforts serves as a significant encouragement to us. We deeply appreciate your constructive feedback, which has immensely contributed to improving the quality of our work. Thanks once again for your invaluable input.

---

### Official Review · Reviewer_Dp49 · 2023-11-03

**Soundness:** 2 fair
**Presentation:** 2 fair
**Contribution:** 2 fair
**Rating:** 6
**Confidence:** 3

**Summary:**

This paper introduces a method that uses a learned reduced-order model to control PDEs. The paper first shows a close connection between the solution from a linear autoencoder and that of dynamic mode decomposition with control. The paper then extends it to the nonlinear domain and introduce a method for control. Experiments demonstrate the effectiveness of the method.

**Strengths:**

Significance: the paper addresses the important problem of modeling and controlling PDEs. The proposed method is effective.

Novelty: the method seems to be novel within the field of neural PDEs. I'm not sure if similar methods (nonlinear autoencoding and control) have been proposed in the robotic control. The component that encourages the model to be exponentially stable is important and novel.

Soundness: the method is demonstrated to outperform the important baseline of deep Koopman and DeepROM.

**Weaknesses:**

Soundness:

For controlling PDEs, it may be important to compare with other classes of baselines, including backpropagation + surrogate model (e.g., [1]), RL-based (e.g., [2]), and predictor + controller methods (e.g., [3]). These are not required, but a more diverse comparison could strengthen the paper as a strong method within the field of PDE control.

[1] Solving PDE-Constrained Control Problems Using Operator Learning, AAAI 2022

[2] Artificial neural networks trained through deep reinforcement learning discover control strategies for active flow control, Journal of fluid mechanics, 2018

[3] Learning to Control PDEs with Differentiable Physics, ICLR 2020

Related works:

There are many neural PDE methods that model the PDE using learned reduced-order models (albeit without control), and may have connections to the proposed method, e.g., [4-8]. The authors are encouraged to state their similarities and differences in the related works section.

[4] Multiscale simulations of complex systems by learning their effective dynamics, Nature Machine Intelligence

[5] Learning to Accelerate Partial Differential Equations via Latent Global Evolution, NeurIPS 2022

[6] Latent space subdivision: stable and controllable time predictions for fluid flow, in Computer Graphics Forum

[7] Model reduction of dynamical systems on nonlinear manifolds using deep convolutional autoencoders,” Journal of Computational Physics

[8] Deep fluids: A generative network for parameterized fluid simulations,” in Computer Graphics Forum

**Questions:**

N/A

---

> ### Author Response · Authors · 2023-11-21
> **Response to reviewer Dp49**
>
> **Q1. For controlling PDEs, it may be important to compare with other classes of baselines, including backpropagation + surrogate model (e.g., [1]), RL-based (e.g., [2]), and predictor + controller methods (e.g., [3]). These are not required, but a more diverse comparison could strengthen the paper as a strong method within the field of PDE control.**
>
> **A1.** We appreciate your suggestion on incorporating diverse comparisons to strengthen the paper. The authors of [1] used backpropagation to optimize the control input of a trained surrogate model through backpropagation, which works well for time-independent PDEs. However, such optimization requires extensive memory and computational resources for time-dependent PDEs when the trajectory is very long, similar to the experiments we used in this paper. For our 1D reaction-diffusion experiment, it is feasible to use this method given the small size of the surrogate model and the low control dimension. However, in experiment, we observed that we can optimize the control input only up to a certain time step. Though the system state reaches the target within this timeframe, it fails to stay stable since the target is an unstable equilibrium and the control input is no longer effective. We have added this result in the supplementary of the revised paper.
>
> As mentioned in [1], RL-based methods (e.g., [2]) require a large number of interactions with the environment and often require system-specific reward functions (e.g., lift and drag of the flow in [2]). In this paper, we investigate a generic scenario and utilize standard distance-based metrics computed from raw states, rather than relying on system-specific rewards or loss functions. Training an RL policy in such a setting is extremely challenging.  Furthermore, RL methods require running numerical solvers in every iteration to provide feedback to the agents, which is computationally expensive. The same concern arises for the methods involving differentiable simulators (e.g., [3]). We agree that it would be interesting to understand the advantages of using a reduced-order model over RL-based methods, particularly without relying on system-specific rewards. We aim to investigate this aspect in future work, focusing on parameters such as training time, sample complexity, and overall performance.
>
>
> **Q2. There are many neural PDE methods that model the PDE using learned reduced-order models (albeit without control), and may have connections to the proposed method, e.g., [4-8]. The authors are encouraged to state their similarities and differences in the related works section.**
>
> **A2.** Thank you for bringing attention to relevant neural PDE methods. We have added similarities and differences with these methods in the related works section.
>
> In [4] and [6], the authors combined encoder-decoder with recurrent networks in latent space to accelerate long-range simulation. The authors of [5] used an autoencoder with a latent evolution model, similar to ours; however, they considered inverse optimization only for static parameters. The authors of [7] used the learned latent representation from an autoencoder to form the trial basis for solving known PDEs using the Galerkin method. In [8], the authors introduced a generative autoencoder with a latent space dynamic model to generate realistic fluid simulation from latent parameters.

---

> > ### Author Response · Authors · 2023-11-22
> > **Gentle reminder**
> >
> > Thanks again for your constructive feedback and comments. We have updated the paper accordingly. As the discussion period ends soon, we would appreciate knowing if our response addressed your concerns.

---

> > > ### Comment · Reviewer_Dp49 · 2023-11-23
> > > **Official Comment**
> > >
> > > Thanks for the updating the paper including relevant works on reduced-order models. The paper is overall sound, novel and interesting. Due to the lack of comparison for other classes of baselines, I remain my score.

---

> > > > ### Author Response · Authors · 2023-11-23
> > > > **Thanks to reviewer Dp49**
> > > >
> > > > Thank you for acknowledging the updated content.  We appreciate your assessment and are pleased that you found the paper sound and novel.

---

### Author Response · Authors · 2023-11-21
**Summary of revision**

We would like to thank the reviewers for their invaluable assessments and insightful suggestions regarding our paper. We deeply appreciate the constructive feedback provided, which guided us to improve the quality of the paper. The changes in the revised paper are highlighted in blue.

Before summarizing the changes made in the revised version, we wish to clarify our paper's contributions. Our intent is not to assert novelty in the concept of autoencoder-based reduced-order models or the utilization of such models for control. Instead, our focus lies in establishing a theoretical connection between traditional and emerging deep learning-based methods for the control of dynamical systems. Our work strategically formulates the architecture and loss functions to serve as a bridge between conventional linear approaches and DNN-based methods, specifically through the lens of a linear autoencoding representation of DMDc.

**Reviewer Dp49** primarily suggested including additional baselines for a more diverse comparison. We report results for an additional baseline that uses backpropagation through the surrogate model.

**Reviewer WjTs** suggested an ablation study with a full-order model and questioned the necessity of reduced-order models. We have added an ablation study on original full-order NI4C which highlights the computational advantage of DeepROM without a substantial compromise in accuracy.

**Reviewer 9vtz** primarily questioned the innovation of our approach. In response, we have clarified our contributions.

**Reviewer Wa1W** suggested adding transformer-based VAE as a baseline. We report results on the prediction performance of such a model for the unforced dynamics.

---

### Meta-Review · Area_Chair_XgYF · 2023-12-11

**Metareview:**

The paper considers a "linear autoencoder" type of objective to learn a latent-variable (or reduced-order) representation of a controlled dynamical system. For the case of linear encoders/decoders and $l_2^2$ loss, the set of optima can be exactly characterized, and resembles the solutions given by DMDc (dynamic mode decomposition with control), which involves truncated SVDs of the data matrix and the augmented data-control matrix. However, the loss is amenable to also using it with non-linear encoders/decoders and some empirical results are provided with the resulting algorithm.

The reviewers were lukewarm about the paper, and I agree. The proofs for the theoretical results are fairly standard and not surprising, in light of classical results that an $l_2^2$ based objective for linear autoencoders is equivalent to calculating a truncation of the SVD (Baldi-Hornik). In essence, $l_2^2$ losses involving matrix products are expected to result in optima that can be described as truncated SVDs of appropriate matrices --- which is what happens here as well, with fairly straightforward calculations. This wouldn't be a problem if the resulting method showed empirical promise: however, as even the authors acknowledge, training difficulties for control systems are challenging, and their method sometimes underperforms baselines.

Ultimately, neither the theory, nor the experiments stand on their own.

**Justification For Why Not Higher Score:**

Ultimately, neither the theory, nor the experiments stand on their own. The theory is fairly standard and follows similar calculations to the ones characterizing optima for linear autoencoders. The experiments are fairly weak and the proposed objective sometimes even underperforms baselines.

**Justification For Why Not Lower Score:**

N/A

---

### Decision · Program_Chairs · 2024-01-16

Reject